# Translational adaptation to heat stress is mediated by RNA 5-methylcytosine in *Caenorhabditis elegans*

Isabela Cunha Navarro[1,2] ID, Francesca Tuorto[3,4,5] ID, David Jordan[1,2], Carine Legrand[3], Jonathan Price[1,2], Fabian Braukmann[1,2], Alan G Hendrick[6], Alper Akay[1,2,7], Annika Kotter[8], Mark Helm[8], Frank Lyko[3] & Eric A Miska[1,2,9,*] ID

## Abstract

Methylation of carbon-5 of cytosines ($m^5C$) is a post-transcriptional nucleotide modification of RNA found in all kingdoms of life. While individual $m^5C$-methyltransferases have been studied, the impact of the global cytosine-5 methylome on development, homeostasis and stress remains unknown. Here, using *Caenorhabditis elegans*, we generated the first organism devoid of $m^5C$ in RNA, demonstrating that this modification is non-essential. Using this genetic tool, we determine the localisation and enzymatic specificity of $m^5C$ sites in the RNome *in vivo*. We find that NSUN-4 acts as a dual rRNA and tRNA methyltransferase in *C. elegans* mitochondria. In agreement with leucine and proline being the most frequently methylated tRNA isoacceptors, loss of $m^5C$ impacts the decoding of some triplets of these two amino acids, leading to reduced translation efficiency. Upon heat stress, $m^5C$ loss leads to ribosome stalling at UUG triplets, the only codon translated by an $m^5C34$-modified tRNA. This leads to reduced translation efficiency of UUG-rich transcripts and impaired fertility, suggesting a role of $m^5C$ tRNA wobble methylation in the adaptation to higher temperatures.

**Keywords** 5-methylcytosine; *Caenorhabditis elegans*; NSUN; RNA modifications; translation efficiency
**Subject Category** RNA Biology
**The EMBO Journal (2021) 40: e105496**

## Introduction

The methylation of carbon-5 of cytosines ($m^5C$) in RNA is a conserved modification in biological systems. $m^5C$ has been detected in tRNAs, rRNAs, mRNAs and non-coding RNAs and is catalysed by $m^5C$ RNA-methyltransferases that utilise SAM as a methyl donor (Liu & Santi, 2000; Boccaletto *et al*, 2017). In humans, RNA $m^5C$ formation is catalysed by the tRNA aspartic acid MTase 1 (TRDMT1/DNMT2), and by seven proteins of the NOP2/Sun domain family (NSUN1-7) (García-Vílchez *et al*, 2019). Pathogenic mutations in humans have been mapped to several genes involved in the $m^5C$ pathway (Abbasi-Moheb *et al*, 2012; Khan *et al*, 2012; Martinez *et al*, 2012; Khosronezhad *et al*, 2015; Komara *et al*, 2015; Ren *et al*, 2015; Haag *et al*, 2016; Nakano *et al*, 2016; Van Haute *et al*, 2016). Despite its conservation and clear relevance, the functions and molecular interactions of the RNA $m^5C$ methylome remain largely unknown. Here, we use *Caenorhabditis elegans* as a model to study the genetic requirements and molecular functions of $m^5C$ modification and its methyltransferases *in vivo*.

$m^5C$ has been implicated in a variety of molecular roles. Among the most highly modified methyltransferase targets are tRNAs and rRNAs, the core components of the translation machinery. tRNAs are methylated by NSUN2, NSUN3, NSUN6 and DNMT2, while rRNAs are methylated by NSUN1, NSUN4 and NSUN5 (García-Vílchez *et al*, 2019).

In some tRNAs, $m^5C$ protects from degradation. Loss of NSUN2-mediated tRNA methylation has been shown to promote cleavage by angiogenin and accumulation of tRNA fragments that interfere with the translation machinery (Tuorto *et al*, 2012; Blanco *et al*, 2014, 2016; Flores *et al*, 2016). Similarly, DNMT2-mediated methylation was found important for protection of tRNAs from stress-induced cleavage in *Drosophila* and mice (Schaefer *et al*, 2010; Tuorto *et al*, 2012, 2015). NSUN6-mediated methylation of tRNAs Cys and Thr promotes a slight enhancement of tRNA thermal stability (Haag *et al*, 2015; Li *et al*, 2018).

In other tRNAs, $m^5C$ modulates translational fidelity. DNMT2 has been shown to facilitate charging of tRNA Asp and

1   Gurdon Institute, University of Cambridge, Cambridge, UK
2   Department of Genetics, University of Cambridge, Cambridge, UK
3   Division of Epigenetics, DKFZ-ZMBH Alliance, German Cancer Research Center, Heidelberg, Germany
4   Division of Biochemistry, Mannheim Institute for Innate Immunoscience (MI3), Medical Faculty Mannheim, Heidelberg University, Mannheim, Germany
5   Center for Molecular Biology of Heidelberg University (ZMBH), DKFZ-ZMBH Alliance, Heidelberg, Germany
6   STORM Therapeutics Limited, Babraham Research Campus, Cambridge, UK
7   School of Biological Sciences, University of East Anglia, Norwich, UK
8   Institute of Pharmacy and Biochemistry, Johannes Gutenberg-University Mainz, Mainz, Germany
9   Wellcome Sanger Institute, Wellcome Genome Campus, Cambridge, UK
    *Corresponding author. Tel: +44 1223 334088; E-mail: eam29@cam.ac.uk

discrimination between Asp and Glu near-cognate codons, thus controlling the synthesis of Asp-rich sequences and promoting translational fidelity (Tuorto *et al*, 2015; Shanmugam *et al*, 2015). NSUN3 methylates exclusively mitochondrial tRNA Met-CAU at the wobble position, which is further modified into f$^5$C by the dioxygenase ALKBH1, facilitating the recognition of AUA and AUG codons as methionine in the mitochondria (Takemoto *et al*, 2009; Van Haute *et al*, 2016; Nakano *et al*, 2016; Haag *et al*, 2016). Lack of f$^5$C affects mitochondrial translation rates in human fibroblasts (Van Haute *et al*, 2016).

The rRNA methyltransferase NSUN1 (yeast *nop2*) methylates C2870 in yeast 25S rRNA. While NSUN1 is an essential gene in all organisms studied thus far, it remains unclear whether this is dependent on its catalytic activity (Sharma *et al*, 2013). Similarly, NSUN4 acts in complex with MTERF4 for assembly of small and large mitochondrial ribosome subunits; however, m$^5$C catalysis does not seem to be essential (Metodiev *et al*, 2014). NSUN5 has been shown to methylate position C2278 in yeast 25S rRNA (Sharma *et al*, 2013; Schosserer *et al*, 2015). Loss of NSUN5-mediated methylation induces conformational changes in the ribosome and modulation of translational fidelity, favouring the recruitment of stress-responsive mRNAs into polysomes and promoting lifespan enhancement in yeast, flies and nematodes. In mice, Nsun5 knockout causes reduced body weight and reduced protein synthesis in several tissues (Schosserer *et al*, 2015; Heissenberger *et al*, 2019).

It remains less clear whether mRNAs are specific targets of m$^5$C-methyltransferases and if m$^5$C is functional in mRNA. Several methods have been used to investigate the presence and function of m$^5$C in coding transcripts and two m$^5$C-binding proteins have been identified so far (Yang *et al*, 2017; Chen *et al*, 2019). However, there is a lack of consensus on the abundance, distribution and relevance of this mark in mRNAs, as the number and identity of putative mRNA m$^5$C sites varies widely between studies (Squires *et al*, 2012; Zhang *et al*, 2012; Tang *et al*, 2015; David *et al*, 2017; Legrand *et al*, 2017; Li *et al*, 2017; Yang *et al*, 2017; Huang *et al*, 2019).

Although previous studies have explored the roles of individual m$^5$C methyltransferases, none have established a systematic dissection of these enzymes as a class, of their specificity, or of their potential molecular and genetic interactions, in any organism. Many questions remain on how the m$^5$C methylome sustains development and normal physiology. In this work, we generated the first mutant animal devoid of any detectable levels of cytosine C5 methylation in RNA, demonstrating that m$^5$C is a non-essential modification under standard conditions. We then used this mutant strain as a genetic tool to map m$^5$C sites onto RNA *in vivo* and determined their impact on translation, development, physiology and stress.

## Results

### m$^5$C and its derivatives are non-essential RNA modifications in *C. elegans*

To identify putative m$^5$C RNA methyltransferases in *C. elegans*, we performed a BLAST analysis and found that the open reading frames *W07E6.1*, *Y48G8AL.5*, *Y39G10AR.21* and *Y53F4B.4* are likely homologues of the human genes NSUN1, NSUN2, NSUN4 and NSUN5, respectively (Fig 1A). Knockdown of these genes through RNAi by feeding revealed that *nsun-1* is an essential gene, as 100% of the animals that had this gene silenced from the first larval stage onwards developed into sterile adults (Fig 1B and C). We could not identify homologous genes of NSUN3, NSUN6, NSUN7 or DNMT2.

m$^5$C RNA methyltransferases utilise two conserved cysteine residues for the methyl group transfer, one of which (TC-Cys) is required for the covalent adduct formation, and the other (PC-Cys) for the release of the substrate following m$^5$C catalysis (King & Redman, 2002). Using CRISPR-Cas9, we introduced mutations converting the TC-Cys into alanine in *nsun-1* (*mj473*), *nsun-2* (*mj458*) and *nsun-4* (*mj457*) (Fig 1D, and Appendix Fig S1A and B). These mutants, as well as a previously reported knockout mutant of *nsun-5* (*tm3898*), are viable and produce viable progeny, suggesting that the individual activity of m$^5$C methyltransferases is not essential for the viability of *C. elegans*. In addition, these results suggest that the essential role played by NSUN-1 in fertility (Fig 1B and C) is independent of the catalytic functions of this protein.

To investigate epistatic interactions among the *nsun* genes in *C. elegans*, we performed genetic crosses between the individual mutants and produced a quadruple mutant in which all *nsun* genes are predicted to be inactive. This strain was viable and fertile and was termed noNSUN. To confirm whether the introduced mutations resulted in catalytic inactivation, and to rule out the existence of additional unknown m$^5$C RNA methyltransferases, we performed mass spectrometry analyses in total RNA from the mutant strains and confirmed that m$^5$C is no longer detectable in this genetic background (Fig 1E; lower limit of detection ~ 0.3 ng/ml, average amount detected in wild type samples 477 ng/ml). We additionally

▶

**Figure 1. m$^5$C and its derivatives are non-essential RNA modifications in *C. elegans*.**

A   Phylogenetic relationship among human and putative nematode NSUN proteins. Unrooted phylogenetic tree of NSUN homologues in *Homo sapiens* and *C. elegans* using entire protein sequence. Phylogenetic tree reconstructed using the maximum likelihood method implemented in the PhyML program (v3.0).

B, C   Knock-down of *nsun* genes through RNAi by feeding. Representative images of wild-type adult animals after silencing of *nsun* genes via RNAi by feeding. Widefield DIC images are 10× magnification (B). Percentage of fertile adults after gene silencing by RNAi (C). *n* = 2 independent experiments, 3 biological replicates each.

D   Mutant alleles used in this study. CRISPR-Cas9 strategy for creation and screening of catalytically inactive alleles of *nsun-1* (*mj473*), *nsun-2* (*mj458*) and *nsun-4* (*mj457*). A homologous recombination template bearing a point mutation to convert the catalytic cysteine into alanine while creating a restriction site for HaeIII was co-injected. For the study of *nsun-5*, a 928 bp deletion allele (*tm3898*) was used. Image not to scale.

E, F   Mass spectrometry quantification of m$^5$C (D) and hm$^5$Cm (E) levels in total RNA from *nsun* mutants. RNA was extracted from populations of L1 animals synchronised by starvation, digested to nucleosides and analysed via LC-MS. *n* = 3 independent biological replicates. n.d. = not detected.

G   Fold change in total RNA modification levels upon loss of m$^5$C. Fold changes were calculated by dividing the peak area ratio of noNSUN samples by the one of wild-type samples. *n* = 3 independent biological replicates. Multiple *t*-tests.

Data information: In (C, E, F, G), data are presented as mean ± SEM. In (C), a representative plot of two independent experiments is shown.

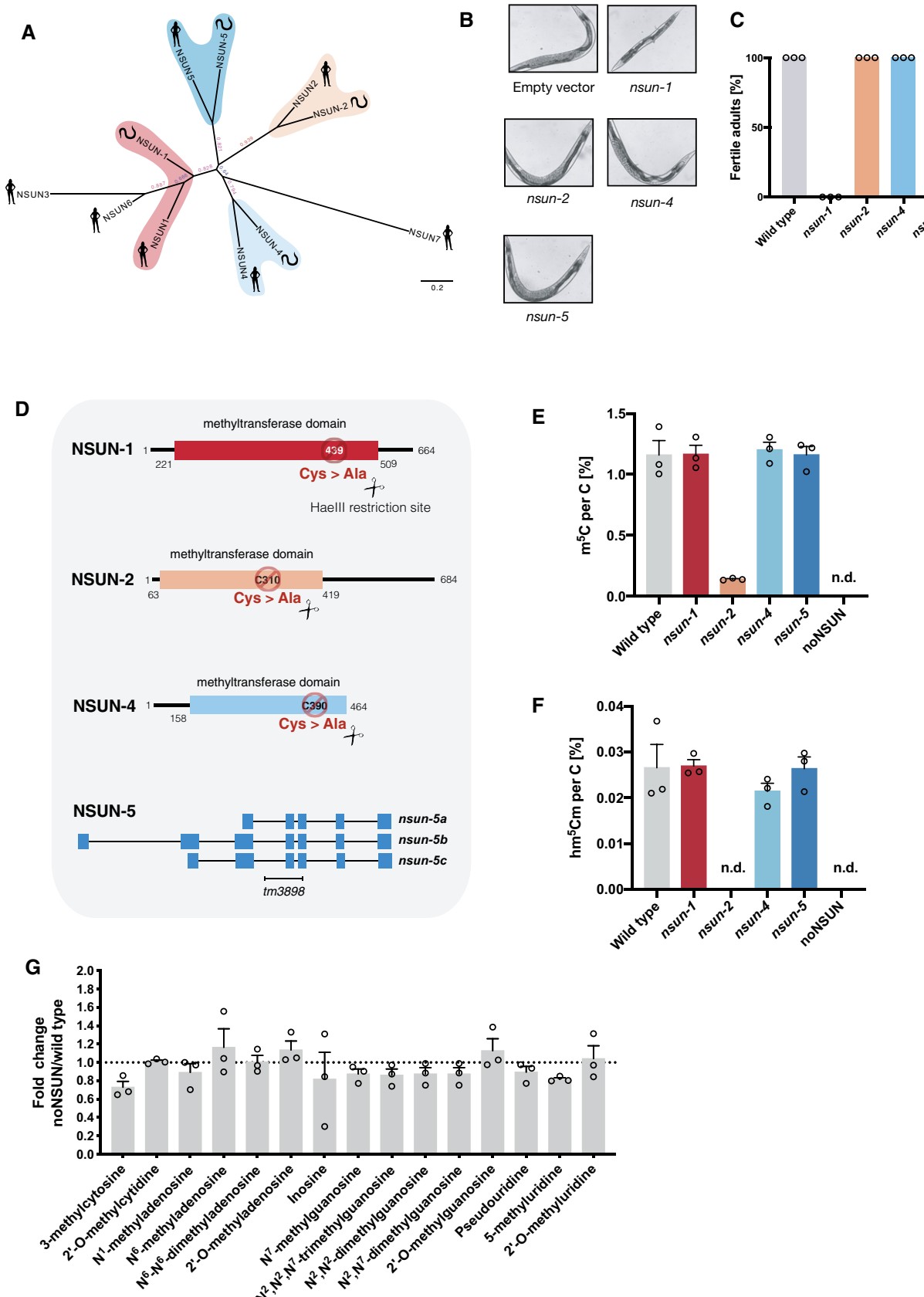

**Figure 1.**

quantified the m⁵C metabolic derivative 2′-O-methyl-5-hydroxy-ymethylcytosine (hm⁵Cm) (Huber *et al*, 2017) and found that this modification is not present either in *nsun-2* or noNSUN mutants (Fig 1F). We therefore conclude that m⁵C and its derivatives are not essential for *C. elegans* viability under laboratory conditions. Furthermore, we showed that NSUN-2 is the main source of m⁵C (88% of total) and that hm⁵Cm sites exclusively derive from NSUN-2 targets in *C. elegans*.

It has been proposed that some RNA modifications may act in a combinatorial manner, providing compensatory effects to each other (Hopper & Phizicky, 2003). This prompted us to investigate whether complete loss of m⁵C would significantly interfere with the levels of other RNA modifications. We performed a mass spectrometry analysis to quantify 15 different modifications in total RNA and found no significant differences between wild-type and noNSUN samples (Fig 1G). Taken together, our data establish the noNSUN strain as a highly specific genetic tool for the study of m⁵C distribution and function *in vivo*.

## The m⁵C methylome of *C. elegans*

Schosserer *et al* demonstrated that position C2381 of 26S rRNA is methylated at carbon-5 by NSUN-5 in *C. elegans*, being involved in lifespan modulation (Schosserer *et al*, 2015). Nevertheless, the m⁵C methylome of this organism remained to be determined. We therefore used the noNSUN strain as a negative control for whole-transcriptome bisulphite sequencing (WTBS) analysis (Legrand *et al*, 2017), aiming to determine the localisation of m⁵C sites in *C. elegans* RNA at single-nucleotide resolution.

We identified C5 methylation at positions C2982 and C2381 of 26S cytoplasmic rRNA and positions C628 and C632 of 18S mito-chondrial rRNA (Fig 2A). Using alignment to rRNA of different organisms, we found that position C2982 is a conserved NSUN1 target (Sharma *et al*, 2013), which has also been recently reported in *C. elegans* (Heissenberger *et al*, 2020). In addition, C2381 has previously been reported as a conserved NSUN5 target (Sharma *et al*, 2013; Schosserer *et al*, 2015). We further confirmed the specificity of these sites using a targeted bisulphite sequencing (BS-seq) approach in individual mutants (Fig EV1A and B). Interestingly, other groups had previously identified adjacent modified sites in mt-rRNA in mice; however, the methylation of only one of the positions was shown to be dependent on NSUN4 activity, while the other was interpreted as a 4-methylcytosine site (Metodiev *et al*, 2014). In the case of *C. elegans*, both positions are NSUN-dependent (Fig 2A).

We found 40 positions to be methylated in stoichiometry higher than 50% in tRNAs, the majority of which being detected in leucine and proline isoacceptors (Fig 2B and C). As anticipated for NSUN2 targeting, modified positions are found in the variable loop region (positions 48, 49, 50), with cytoplasmic tRNA Leu-CAA carrying an additional modification at the wobble position (C34) (Blanco *et al*, 2014; Burgess *et al*, 2015) (Fig 2C). Using targeted BS-seq, we demonstrated that NSUN-2 is indeed responsible for both C34 and C48 methylation in tRNA-Leu (Fig EV1C). In agreement with the lack of homologous genes of DNMT2 and NSUN6 in *C. elegans*, no methylation was found on conserved RNA targets of these enzymes (Goll *et al*, 2006; Schaefer *et al*, 2010; Long *et al*, 2016; Fig 2C).

Contrasting with what was observed for tRNAs and rRNAs, non-conversion of mRNA sites in *C. elegans* is rare and occurs at much lower stoichiometry. Lowering the non-conversion threshold from 50 to 25–40%, we detected 188 positions that remained unconverted after bisulphite treatment exclusively in wild-type samples, i.e. putative m⁵C sites (Fig 2D, *x*-axis). Using the same thresholds to probe for likely artefacts revealed 88 positions that remained unconverted exclusively in noNSUN samples, i.e. non-conversion was only 46% less frequent (Fig 2D, *y*-axis). It is also noteworthy that positions remaining reproducibly highly unconverted equally in wild-type and noNSUN samples are more frequent in mRNAs, when compared to tRNAs and rRNAs (Fig 2A, B and D). In summary, we found no evidence of a widespread distribution of m⁵C in coding transcripts. To investigate whether the presence of common characteristics could support a subset of the aforementioned 188 positions as *bona fide* methylated sites, we performed gene ontology, motif search, genomic localisation and secondary structure analyses on these transcripts and sites; however, no significant shared features were found. While our data do not completely rule out the existence of m⁵C methylation in mRNAs, it demonstrates that this mark cannot be detected in high stoichiometry in *C. elegans*, as observed in tRNAs and rRNAs.

Finally, we attempted to identify m⁵C sites in small RNAs in our data set. Given that the fractionation used in this protocol aimed to enrich for tRNAs (60–80 nt), we could not detect microRNA reads in abundance for confident analysis. Nevertheless, we found high NSUN-dependent non-conversion rates (> 80%) in five non-coding RNAs (approximately 60 nt long) previously identified in *C. elegans* (Lu *et al*, 2011; Xiao *et al*, 2012) (Fig 2E). Secondary structure predictions suggest that the methylated sites are often found on the base of a stem-loop, reminiscent of tRNA variable loops (Fig EV2). Further experiments will be required to determine the functionality of these m⁵C sites.

---

**Figure 2. The m⁵C methylome of *C. elegans*.**

A, B   Site-specific methylation analysis by whole-transcriptome bisulphite sequencing. Scatter plots show individual cytosines and their respective non-conversion rates in rRNAs (A) and tRNAs (B) of wild-type and noNSUN strains; pie chart showing most frequently methylated tRNA isoacceptors.

C   Heat map showing non-conversion rates of tRNA positions methylated in stoichiometry higher than 50% and of tRNA positions predicted to be targets of DNMT2 and NSUN6.

D   Site-specific methylation analysis by whole-transcriptome bisulphite sequencing. Scatter plot shows density of cytosines and their respective non-conversion rates in mRNAs of wild-type and noNSUN strains.

E   Heat map showing non-conversion rates of small non-coding RNA positions methylated in stoichiometry higher than 50%.

Data information: In (A–E), *n* = 3 independent biological replicates.

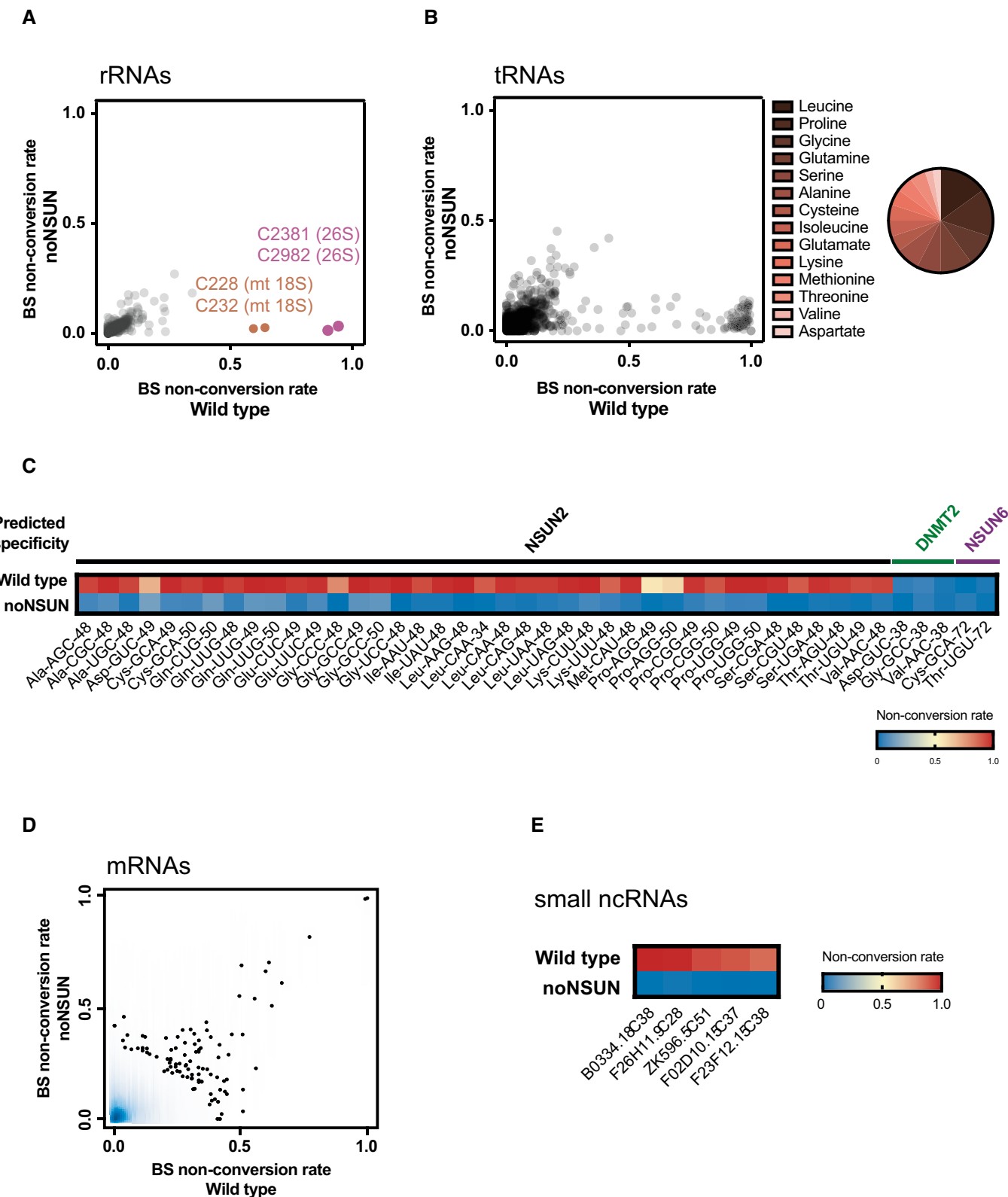

**Figure 2.**

**NSUN-4 is a multisite-specific tRNA/rRNA-methyltransferase in the mitochondria of *C. elegans***

Interestingly, we found the mitochondrial tRNA Met-CAU to be methylated at a very high rate (94.7%) (Fig 3A). Supporting our finding, previous articles also indicated the detection of this modified site in *Ascaris suum* and *C. elegans* (Watanabe *et al*, 1994; Nakano *et al*, 2016). This was unexpected, as previous reports have shown that this position is methylated by NSUN3 (Haag *et al*, 2016; Nakano *et al*, 2016; Van Haute *et al*, 2016). As *C. elegans* does not have an NSUN3 homologue, this implies that mitochondrial tRNAs can be modified by alternative enzymes. A BLAST analysis of the human NSUN3 methyltransferase domain against the *C. elegans* proteome showed higher similarity to NSUN-4 (30% identity), followed by NSUN-2 (26% identity). Moreover, we observed that, among human NSUN genes, NSUN3 and NSUN4 share the highest percentage of similarity (Fig 3B). Using a targeted BS-seq approach, we probed the methylation status of position C34 in mitochondrial tRNA Met-CAU from wild-type, *nsun-2* and *nsun-4* strains. Our results indicate that NSUN-4 is responsible for the catalysis of m$^5$C in this position in *C. elegans* (Fig 3C).

NSUN-4 is the only mitochondrial rRNA m$^5$C methyltransferase identified to date. To confirm that the previously reported role of NSUN4 is also conserved in *C. elegans*, we performed targeted BS-seq in 18S mitochondrial rRNA and found that methylation of positions C628 and C632, as well as C631, is mediated by NSUN-4 (Fig 3D). Notably, methylation of position C631 was also detected by WTBS, however in reduced stoichiometry (23.5% in wild type vs. 0.04% in noNSUN). Taken together, our results show that NSUN-4 is a multisite-specific tRNA/rRNA mitochondrial methyltransferase in *C. elegans*.

To investigate when the divergence of NSUN3 arose in evolution, a phylogenetic analysis of NSUN3 was performed using Treefam and, given the high sequence similarity, NSUN3 and NSUN4 sequences were automatically included in the generated cladogram. While *Drosophila* and *C. elegans* only have NSUN4, vertebrate model organisms as basal as zebrafish have both NSUN3 and NSUN4 (Fig 3E). A more expanded version of the tree indicates the presence of NSUN4, but not NSUN3, in sea lampreys (http://www. treefam.org/family/TF321304#tabview=tab1), suggesting that NSUN3 diverged from NSUN4 in vertebrates.

**Loss of m$^5$C leads to temperature-sensitive reproductive phenotypes**

The individual or collective introduction of mutations in *nsun* genes failed to induce noticeable abnormal phenotypes. We therefore performed a more extensive characterisation of the mutant strains using a live imaging-based phenotypic analysis (Akay *et al*, 2019). As a proxy for reduced fitness, we chose to analyse the number of viable progeny and occurrence of developmental delay (growth rate, as measured by body length). In comparison with wild-type animals, we observed a delay in all mutant strains, which persists throughout development and into adulthood. This difference is greater in noNSUN animals, especially as this strain transitions from L4 stage to young adulthood (Fig 4A). When comparing mutants' sizes at young adult stage at 20°C, noNSUN animals are, on average, five times smaller than wild type (Fig 4B). While this difference reflects a developmental delay, noNSUN animals remain 20% smaller even when they reach adulthood themselves (Fig 4B). In addition, the noNSUN strain shows a 25% reduction in brood size, which is comparable to what is observed in *nsun-1* and *nsun-5* individual mutants (Fig 4C).

*Caenorhabditis elegans* stocks can be well maintained between 16 and 25°C, being most typically kept at 20°C. To gain insights into how the loss of m$^5$C impacts development under different environmental conditions, wild-type and noNSUN animals were cultured at 25°C for three generations and subjected to automated measurements. As shown in Fig 4D, the reproductive phenotype previously observed in the noNSUN strain (Fig 4C) is significantly aggravated at this temperature. This suggests that the phenotypes arising from loss of m$^5$C are temperature-sensitive, pointing towards an involvement of this modification in the adaptation to environmental changes.

**Loss of m$^5$C impacts translation efficiency of leucine and proline codons**

To explore the impact of temperature stress in the absence of m$^5$C while avoiding the confounding effect introduced by differences in brood size, we performed further experiments using an acute heat shock treatment. To investigate whether the observed phenotypes are linked to abnormalities in protein translation rates, we quantified the polysomal fraction in wild-type and noNSUN adult animals subject to heat shock at 27°C for 4 h and found no significant differences (Fig 5A, Appendix Fig S2).

To gain insights into transcriptional and translational differences resulting from the loss of m$^5$C, we performed transcriptomic and ribosome profiling analyses. The latter allows the quantification of active translation by deep-sequencing of the mRNA fragments that are protected from nuclease digestion by the presence of ribosomes (ribosome-protected fragments—RPFs). RPFs showed the expected 3 nt periodicity along the coding domain sequences of mRNA, with

---

**Figure 3.  NSUN-4 is a dual tRNA/rRNA methyltransferase in *C. elegans*.**

A    RNA bisulphite sequencing map for mitochondrial tRNA Met-CAU in wild-type (top) and noNSUN (bottom) strains. Each row represents one sequence read, and each column one cytosine.
B    Per cent identity matrix of human NSUN proteins according to the Clustal Omega multiple alignment tool.
C, D    Targeted bisulphite-sequencing heat map showing non-conversion rates of cytosines in mitochondrial tRNA Met-CAU (C) and mitochondrial 18S rRNA (D). Each row represents one genetic strain analysed, and each column represents one cytosine.
E    Treefam phylogenetic tree based on sequence conservation of NSUN3 proteins in different model organisms. Bootstrap values are indicated on branches.

Data information: In (A), a representative map of the replicates is shown, *n* = 3 independent biological replicates. In (C, D), the average of two experiments is plotted, *n* = 2 independent biological replicates, 10 clones sequenced per strain, per replicate. Similar effects were observed in all replicates analysed.

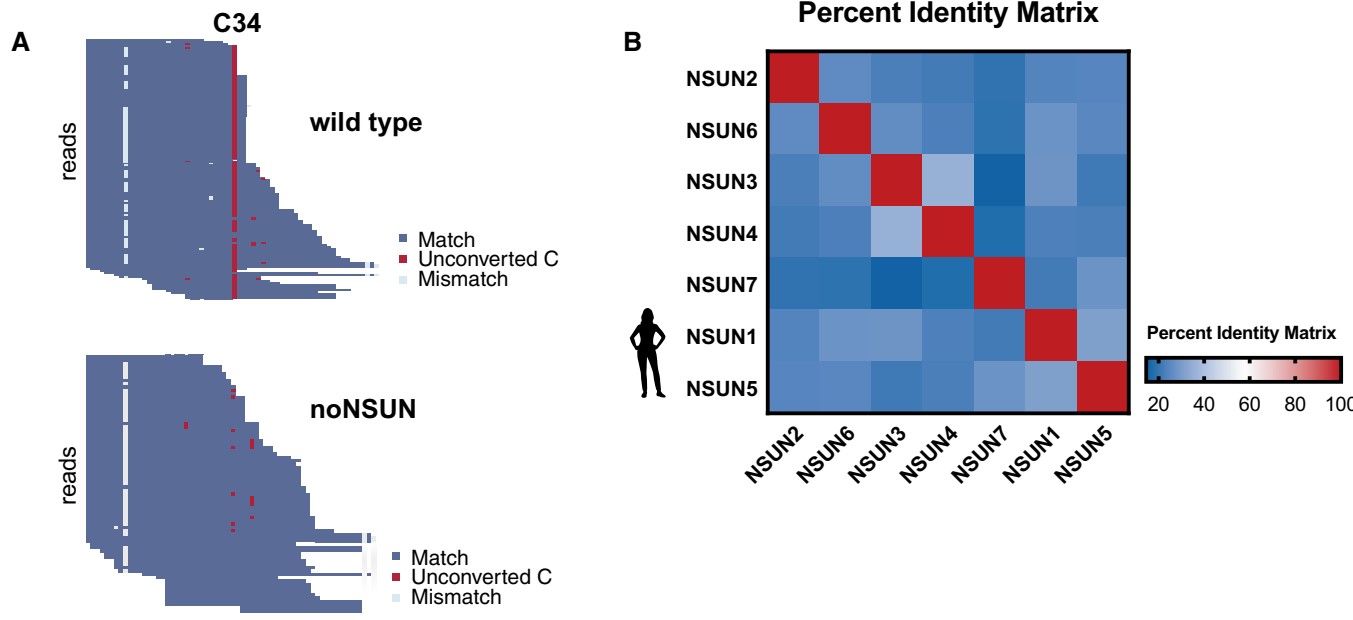

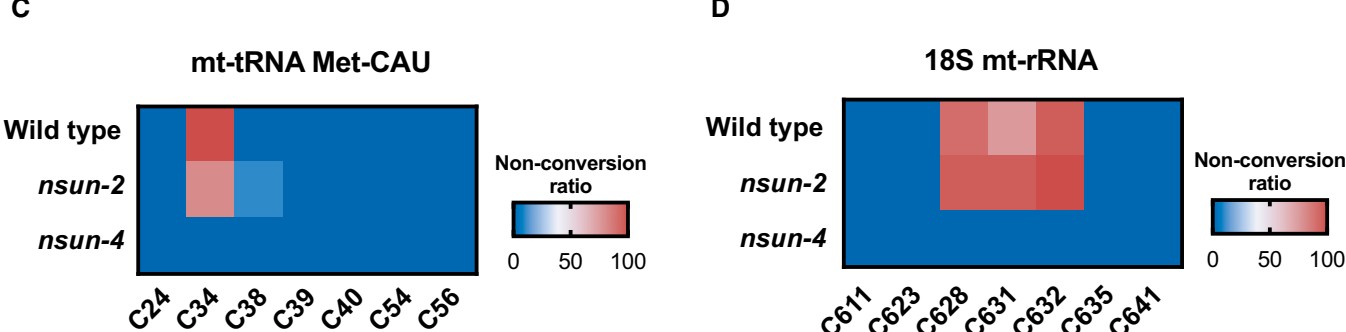

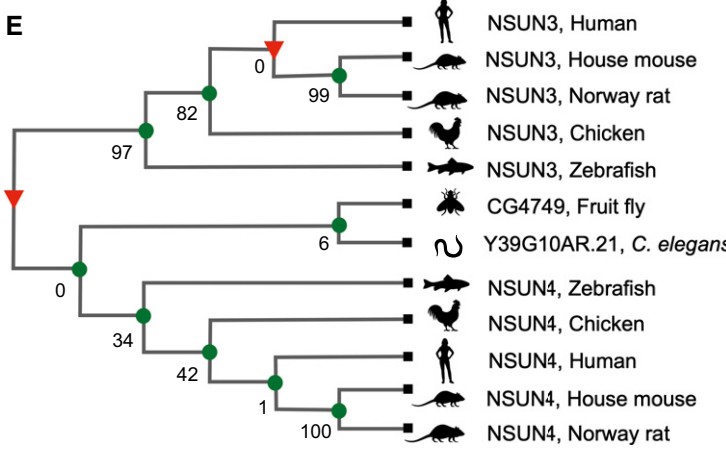

**Figure 3.**

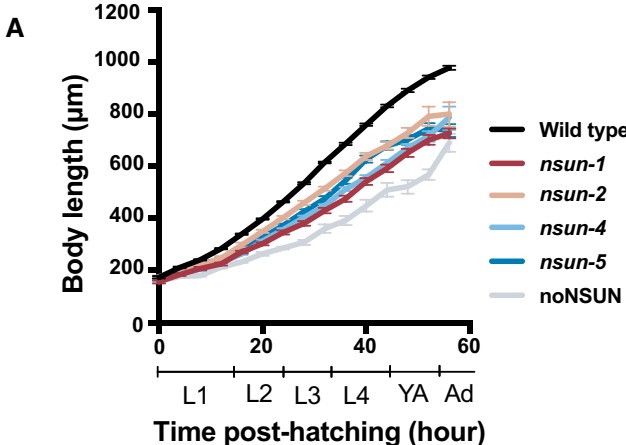

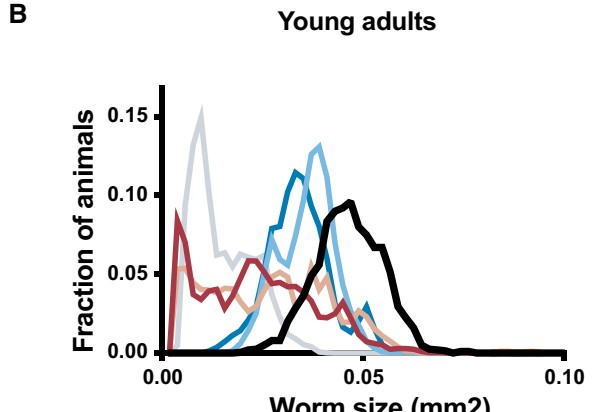
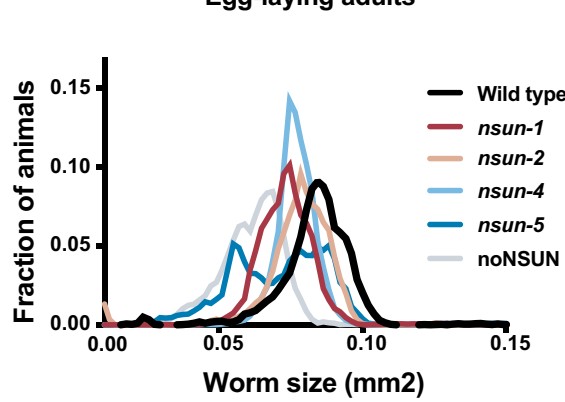

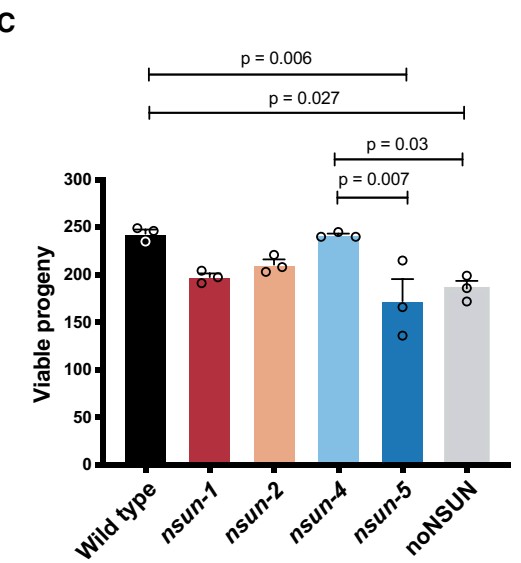
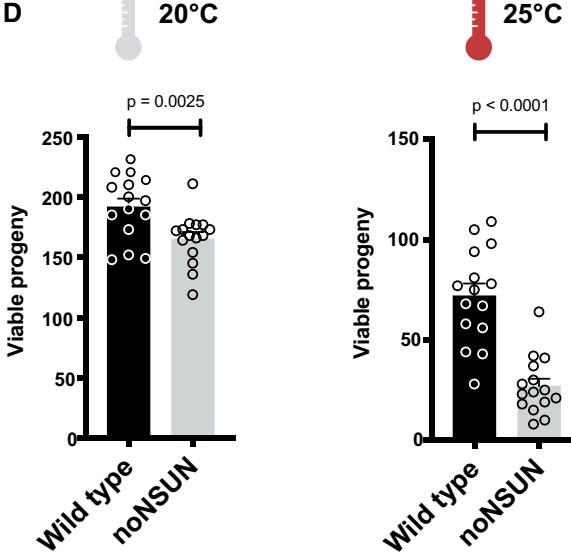

Figure 4.

**Figure 4.  Loss of m5C leads to a temperature-sensitive reproductive phenotype.**

A    Body length of individual *nsun* mutants throughout development (*n* = 44, 7, 7, 7, 8, 8) in ~ 4 h windows. L1-L4 refers to the larval stages, YA and Ad to young adult and adult, respectively.

B    Size of mutant *nsun* strains at young adult and egg-laying stages. Approximately 100 synchronised young adults of each strain were measured. Images were processed using custom algorithms to recognise *C. elegans* and measure their cross-sectional area.

C, D    Viable progeny counts of wild-type and *nsun* mutant strains at 20°C (C) and of wild-type and noNSUN strains at 20°C and 25°C (D). Automatic counting was done using a MATLAB script which processed plate images in real-time.

Data information: In (A), error bars indicate the 95% confidence interval of the median. In (C), data are presented as mean $\pm$ SEM, one-way ANOVA. In (D), data are presented as mean $\pm$ SEM, unpaired two-tailed Student's *t*-test.

the majority of reads in frame (Appendix Fig S3A and B). Furthermore, RNA-seq and Ribo-seq counts of genes showed high correlation, and variation in the gene counts could be attributed to the difference in samples analysed (Appendix Fig S3C–E). Loss of m5C did not greatly impact the nature of the heat stress response, as most differentially transcribed and translated genes upon heat stimulus showed agreement, or very subtle differences between wild-type and noNSUN strains (Appendix Fig S4). We found that differentially transcribed genes upon loss of m5C are mainly involved in cuticle development (Fig EV3A), while differentially translated genes are enriched in components of the cuticle and ribosomes, as well as RNA-binding proteins (Fig EV3B).

We then evaluated genome-wide codon occupancy during translation elongation in both temperatures and found that loss of m5C leads to increased ribosome occupancy at leucine and proline codons. Upon heat shock, Leu-UUG codons showed the highest ribosome density observed in the noNSUN strain, suggesting that translation of this codon is slowed during heat stress in the absence of m5C (Figs 5B and EV4A). We investigated this phenomenon more closely in different transcripts and found that ribosome stalling at UUG codons seems to be context-dependent, as it only occurs in a small subset of UUG codons (Figs 5C and EV4B). Interestingly, as shown in our WTBS analysis, leucine and proline are the most frequently methylated tRNA isoacceptors in *C. elegans* (Fig 2B). In addition, tRNA Leu-CAA, responsible for decoding of UUG codons, is the only cytoplasmic tRNA bearing an m5C-modified wobble position (Fig 2C).

As a downstream consequence of ribosome stalling, we found translation efficiency of UUG-, leucine- and proline-rich genes to be significantly reduced in the noNSUN strain. While this effect can be observed in both temperatures in leucine-rich transcripts and at 20°C in proline-rich ones, it occurs in a heat shock-dependent manner in UUG-rich transcripts, suggesting an involvement of m5C wobble methylation in the adaptation to heat stress (Fig 5D). Finally, we found that translation efficiency is further reduced as transcripts get more enriched in the affected codons (Fig EV5).

## Discussion

Chemical modifications of RNA occur in organisms from all kingdoms of life and are often highly conserved throughout evolution, as is the case of the methylation of carbon-5 in cytosines (Huber *et al*, 2015; Boccaletto *et al*, 2017). Despite that, there is a growing body of evidence showing that several RNA modifications are individually not required for development under controlled conditions (O'Connor *et al*, 2018, reviewed in Sharma & Lafontaine, 2015 and Hopper & Phizicky, 2003). Our results reignite a recurrent question in the epitranscriptomics field: why are so many of these chemical marks extensively conserved throughout evolution and, yet, organisms often present subtle phenotypes in their absence? Ribonucleoside modifications occur in an overwhelming diversity and, in some cases, might (i) exert subtle molecular effects, (ii) act in a combinatorial or redundant manner with other modifications or (iii) be the result of relaxed enzymatic specificity (Phizicky & Alfonzo, 2010; Jackman & Alfonzo, 2013).

While the absence of m5C in RNA did not give rise to overt phenotypes under standard laboratory conditions, a more detailed analysis of the mutants revealed developmental and fertility defects. Previous studies have shown that levels of several RNA modifications, including m5C, are responsive and can react dynamically to a wide range of environmental challenges, such as toxicants, starvation and heat shock, thus potentially supporting organismal adaptation (Chan *et al*, 2010; van Delft *et al*, 2017). In agreement with this idea, we observed a temperature-dependent aggravation of reproductive phenotypes in m5C-deficient *C. elegans*. In nature, where the environmental conditions vary greatly, such genotypes would likely be selected against in a wild population.

Our results suggest an RNA methylation-independent essential role for NSUN-1 in germline development. Consistently, Nop2p/Nol1/NSUN1 has been shown to be an essential gene in yeast, mice and *Arabidopsis* (De Beus *et al*, 1994; Sharma *et al*, 2013; Burgess *et al*, 2015; Kosi *et al*, 2015). In *Saccharomyces cerevisiae*, both depletion and catalytic mutation of Nop2p lead to lower levels of

**Figure 5.  Loss of m5C impacts translation efficiency of leucine and proline codons.**

A    Fraction of polysomal ribosomes quantified from polysome profiles in the wild-type and noNSUN strains subject to a 4 h heat shock at 27°C. ns = non-significant.

B    Heat map showing P-site codon occupancy according to the colour scale at 20 and 27°C. Red and blue refer to enhanced and reduced codon occupancy, respectively, in the noNSUN strain relative to wild type. Leucine and proline codons are marked in red.

C    Ribosome-protected fragment (RPF) counts in each sample plotted along *ife-1* and *pat-10* CDS. Vertical grey lines indicate UUG codons.

D    Translation efficiency of UUG-enriched, leucine-enriched, proline-enriched and random genes in each sample. A gene was considered enriched in a certain codon when the proportion of this codon in the gene was at least 3-fold higher than the proportion of the same codon across the transcriptome.

Data information: In (A–E), *n* = 3 biological replicates. In (A), data are presented as mean $\pm$ SEM, unpaired two-tailed Student's *t*-test. In (D), box plots show the median (central band) and IQR (boxes) $\pm$ 1.5 $\times$ IQR (whiskers), Welch's *t*-test, *P*-value < 0.05.

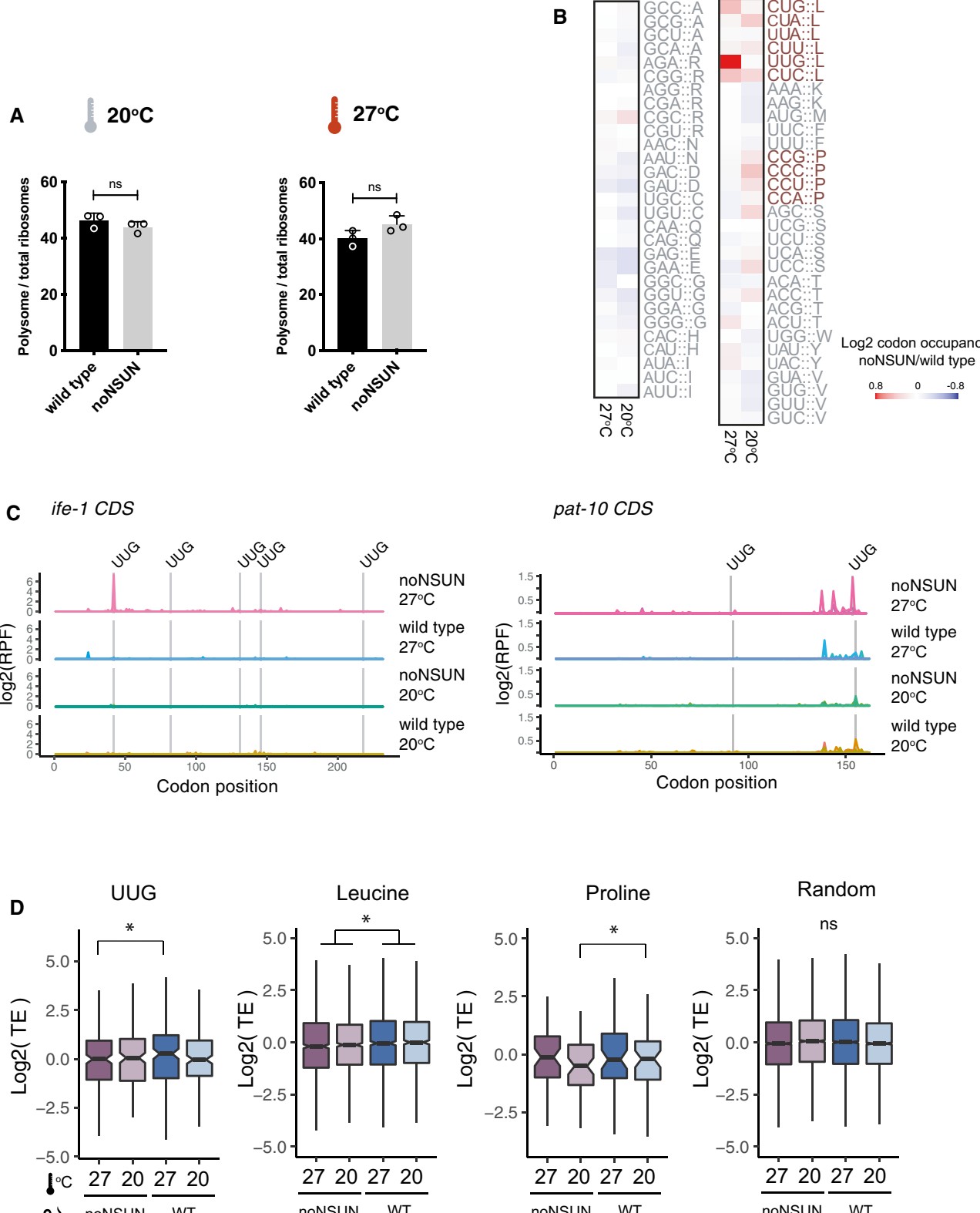

**Figure 5.**

60S ribosomal subunits, supporting the idea that reduced methylation affects rRNA processing and translation (Hong *et al*, 1997; Hong *et al*, 2001; Sharma *et al*, 2013). In contrast, Bourgeois *et al* (2015) reported that loss of Nop2p-mediated m[5]C had no effect on ribosome synthesis and phenotype. A similar phenomenon has been observed for NSUN4 in mice, as well as for Dim1 and Trmt12 in yeast, and their human homologues DIMT1L and WBSCR22, where the presence of the enzyme, rather than its catalytic activity, is required for viability (Lafontaine *et al*, 1995; Metodiev *et al*, 2014; Zorbas *et al*, 2015). It has been proposed that the essential binding of certain rRNA methyltransferases represents a quality control step in ribosome biogenesis, committing rRNA to methylation during the maturation process (Lafontaine *et al*, 1998).

Taking advantage of the noNSUN strain as a tool to increase the confidence of WTBS analysis, we produced the first comprehensive list of m[5]C sites throughout *C. elegans* transcriptome. Using a targeted approach, we showed that NSUN-4 has both rRNA and tRNA targeting capabilities in the mitochondria. It has been suggested that binding of MTERF4 and NSUN4 in a complex is responsible for targeting the methyltransferase to rRNA in the mitochondria (Spåhr *et al*, 2012; Yakubovskaya *et al*, 2012; Metodiev *et al*, 2014). Nevertheless, genetic evidence suggests that NSUN4 methylates rRNA independently of MTERF4 in mice (Metodiev *et al*, 2014). *C. elegans* has an MTERF4 homologue (K11D2.5), and most residues involved in the interaction with NSUN4 appear conserved, suggesting that a similar interaction with this co-factor could occur (Spåhr *et al*, 2012).

In humans, NSUN3-mediated methylation at position 34 of mitochondrial tRNA Met-CAU is further modified by the dioxygenase ALKBH1 to form f[5]C (Haag *et al*, 2016; Nakano *et al*, 2016). Previous studies explored differential methods for the detection of f[5]C, indicating that 35–100% of tRNA Met-CAU molecules are f[5]C-modified, while the whole population is at least m[5]C-modified (Haag *et al*, 2016; Van Haute *et al*, 2016; Kawarada *et al*, 2017). As f[5]C reacts as an unmodified cytosine upon sodium bisulphite treatment, it was surprising to detect high levels of non-conversion (95%) in our study. Nakano *et al* (2016) have used DNA probes in reciprocal circulating chromatography followed by mass spectrometry to demonstrate high stoichiometry of f[5]C in this site in *C. elegans* RNA. In addition, an ALKBH1 homologue (Y51H7C.5) has recently been discovered and implicated in mitochondrial protein biogenesis in the nematode (Wagner *et al*, 2019). These results support the existence of a f[5]C pathway in *C. elegans*. However, as we used a sequencing-based method that does not discriminate between precursors or mature tRNAs, it is possible that our method detects mainly primary transcripts or precursor molecules, which have not been oxidised by ALKBH1.

Using the noNSUN strain as a negative control, we investigated the presence of m[5]C in coding transcripts. Several reports have shown that m[5]C is a common mRNA modification (Squires *et al*, 2012; Amort *et al*, 2017; David *et al*, 2017; Yang *et al*, 2017). However, results derived from BS-seq can be influenced by several factors, such as incomplete deamination, protection due to secondary structures, presence of other modifications, protein binding and sequencing errors, among others (summarised in Legrand *et al*, 2017). Given these technical drawbacks, the noNSUN strain represented an unprecedentedly stringent negative control, which allowed for exclusive detection of highly specific methylation.

Despite the detection of positions with 20–30% NSUN-dependent non-conversion, we detected a similar number of positions with NSUN-independent non-conversion at these rates, which we interpret as false positives. This poses a statistical challenge on the interpretation of such non-converted positions as methylated. Our main conclusion, therefore, is that the data do not provide evidence for widespread or high stoichiometry m[5]C methylation of coding transcripts in *C. elegans*. This is in agreement with earlier work using chromatography (Desrosiers *et al*, 1974; Adams & Cory, 1975; Salditt-Georgieff *et al*, 1976) and other reports that have detected very few or no m[5]C sites in eukaryotes by BS-seq (Edelheit *et al*, 2013; Khoddami & Cairns, 2013; Legrand *et al*, 2017; Khoddami *et al*, 2019).

Using ribosome profiling, we investigated the genome-wide effect of loss of m[5]C in translational speed and efficiency and found leucine and proline translation to be affected. The strongest effect by far was observed in a heat shock-dependent manner in UUG codons, which rely on the only tRNA modified at the wobble position—tRNA Leu-CAA. Chan *et al* (2012) found that m[5]C level specifically at position 34 of tRNA Leu-CAA is upregulated upon oxidative stress in yeast. The presence of this modification was shown to enhance translation efficiency of a UUG-rich luciferase reporter construct, as *trm4Δ* (NSUN2 homologue mutant) cells showed significantly lower levels of reporter activity, especially under oxidative stress. The biological relevance of these findings was linked to an abnormally high frequency of UUG codons in transcripts of specific ribosomal protein paralogues (Chan *et al*, 2012).

In summary, m[5]C supports *C. elegans* fitness at higher temperatures and enhances the translational efficiency of leucine and proline codons in physiology and stress. Our work highlights a specific role of cytosine C5 methylation in facilitating translation of leucine UUG codons upon heat shock, suggesting that m[5]C tRNA wobble methylation is involved in the adaptation to heat stress.

# Materials and Methods

### Genetics

*Caenorhabditis elegans* strains were grown and maintained as described in Brenner (Brenner, 1974). The strains were kept at 20°C, unless otherwise indicated. HB101 strain *Escherichia coli* was used as food source (*Caenorhabditis* Genetics Center, University of Minnesota, Twin Cities, MN, USA). Bristol N2 was used as the wild type strain.

### Gene silencing by RNAi

Empty vector, *nsun-1* (*W07E6.1*), *nsun-2* (*Y48G8AL.5*), *nsun-4* (*Y39G10AR.21*) and *nsun-5* (*Y53F4B.4*) bacterial feeding clones were kindly provided by Prof. Julie Ahringer's laboratory (Kamath & Ahringer, 2003). Single colonies were inoculated in LB-Ampicillin 100 μg/ml and cultured for 8 h at 37°C. Bacterial cultures were seeded onto 50 mm NGM agar plates containing 1 mM IPTG and 25 μg/ml Carbenicillin at a volume of 200 μl of bacterial culture per plate, and left to dry for 48 h. Fifty synchronised L1 larvae were placed onto RNAi plates and left to grow until adult stage. Adults were scored for fertility (presence of embryos in the germline).

## CRISPR-Cas9 gene editing

CRISPR-Cas9 gene editing was performed as in Paix et al (2015). Briefly, injection mixes were prepared in 5 mM Tris pH 7.5 as follows: 20 μg of tracrRNA (Dharmacon), 3.2 μg of dpy-10 crRNA (Dharmacon), 200 ng of dpy-10 homologous recombination template (Sigma Aldrich), 8 μg of target gene gRNA (Dharmacon), 1.65 μg of homologous recombination template (Sigma), up to a volume of 11.5 μl. The mix was added to 10 μg of Cas9 (Dharmacon) to a final volume of 15 μl and incubated at 37°C for 15 min. For the creation of nsun catalytic mutants, a homologous recombination template bearing a point mutation to convert the catalytic cysteine into alanine while creating a restriction site for HaeIII was co-injected. Following incubation, the mix was immediately microinjected into the germline of N2 young adults. After injection, animals were left to recover in M9 medium, then transferred to individual plates and left to recover overnight at 20°C. Successful injections led to the hatch of dumpy and roller animals. From positive plates, 96 animals were individualised for self-fertilisation and genotyped for the relevant alleles. Same process was performed with F2s, until a homozygous population was isolated. Each strain was backcrossed at least three times with the wild type strain.

## RNA extraction (mass spectrometry and WTBS)

The strains of interest (N2 and noNSUN) were grown in 90 mm plates until gravid adult stage, washed three times with M9 and pelleted by centrifugation at 775 g for 2 min. Gravid adults were resuspended in 4 ml of bleaching solution (final concentration 177 mM NaOH, 177 mM NaOCl solution—free chlorine 4–5%) and vortexed vigorously for 7 min. Recovered embryos were washed four times to remove any traces of bleach and left to hatch in ml of M9 for 24 h at 20°C in a rotating wheel. Synchronised L1 starved larvae were used for RNA extraction. Independent triplicates were obtained from three different generations.

Nematodes were washed thoroughly in M9 to remove bacterial residue and pelleted in RNAse-free tubes at 775 g for 2 min. 500–1,000 μl of TRIsure (Bioline) and 100 μl of zirconia beads were added, and the samples were subjected to three cycles of 4000 g with 20 s breaks on Precellys to crack open the animals. 100 μl of chloroform was added to the tubes, which were then shaken vigorously for 15 s and incubated at room temperature for 3 min. Samples were centrifuged at 12,000 g for 15 min at 4°C, and the aqueous phase of the mixture was carefully recovered and transferred to a fresh RNAse-free tube. RNA was precipitated with 500 μl of cold isopropanol at room temperature for 10 min and then centrifuged at 12,000 g for 15 min at 4°C. The supernatant was carefully removed, the pellet was washed and vortexed with 1 ml of 75% ethanol and centrifuged at 7,500 g for 5 min at 4°C. RNA pellet was air-dried, dissolved in the appropriate volume of DEPC-treated water and the concentration, and 260/280 and 260/230 ratios were measured by Nanodrop. RNA integrity was evaluated in the Agilent 2200 Tapestation system.

## RNA mass spectrometry

Up to 10 μg of RNA was digested by adding 1 μl digestion enzyme mix per well in a digestion buffer (4 mM Tris–HCl pH 8, 5 mM MgCl$_2$, 20 mM NaCl) in a total volume of up to 100 μl. The digestion enzyme mix was made by mixing benzonase (250 U/μl, Sigma Aldrich), phosphodiesterase I from Crotalus adamanteus venom (10 mU/μl, Sigma Aldrich) and Antarctic phosphatase (5 U/μl, NEB) in a ratio of 1:10:20. The reaction was incubated overnight at 37°C. The following day, an equal volume of $^{13}$C, $^{15}$N-labelled uridine (internal control, previously dephosphorylated; Sigma Aldrich) in 0.1% formic acid was added to each reaction and this was subsequently prepared for LC-MS-MS by filtration through 30 kDa molecular weight cut-off filters (Sigma).

Samples were resolved using a Thermo Scientific U3000 UPLC system on a gradient of 2–98% (0.1% formic acid/acetonitrile) through an Acquity 100 × 2.1 mm C-18 HSS T3 column and analysed on a QExactive-HF Orbitrap High Resolution Mass Spectrometer (Thermo Fisher Scientific, IQLAAEGAAPFALGMBFZ) in positive full-scan mode, and the results were deconvoluted using the accompanying Xcalibur Software. Nucleosides of interest were identified by both retention times and accurate masses, compared with purified standards and quantified accordingly.

## Whole transcriptome bisulphite sequencing

Bisulphite sequencing experiments were performed as previously described in Legrand et al (2017). RNA was fractionated into < 200 nt and > 200 nt using a modified mirVana miRNA isolation kit (AM1560) protocol. Briefly, 50 μg of RNA in a volume of 80 μl were mixed with 400 μl of mirVana lysis/binding buffer and 48 μl of mirVana homogenate buffer and incubated for 5 min at room temperature. Next, 1/3 volume (176 μl) of 100% ethanol was added and thoroughly mixed by inversion, and the mixture was incubated for 20 min at room temperature. After addition of 0.8 μg of Glycoblue, the samples were spun down at 2,500 g for 8 min at 21°C for precipitation of long RNAs. The supernatant containing the short fraction was transferred to a fresh tube, and the RNA pellet was washed in 1 ml of cold 75% ethanol before centrifugation at maximum speed for at least 20 min at 4°C. The pellet was finally air-dried and resuspended in DEPC-treated water. For short fraction RNA precipitation, 800 μl of isopropanol was added to the supernatant and the mixture was incubated at −80°C for at least 20 min. Next, 20 μg of Glycoblue was added and the mixture was spun down at maximum speed for at least 20 min at 4°C. The pellet was washed with cold 70% ethanol and air-dried before resuspension in DEPC-treated water. Depletion of ribosomal RNA was performed on the short fractions and on half of the long fractions using a Ribozero rRNA removal kit (Illumina), according to the supplier's instructions. The other half of long fractions was processed as Ribo + samples. RNA was stored at −80°C until the moment of use.

The long fractions (with and without rRNA depletion) were further processed with the NEBNext Magnesium RNA Fragmentation Module (NEB), as described in the manual. 3 min of fragmentation at 94°C has been established to lead to a peak at approximately 250 nt, appropriate for the final 100 bp paired-end sequencing. The fragmented RNA was precipitated using ethanol with 20 μg GlycoBlue at −80°C for at least 10 min.

Samples were treated with TURBO DNase (Ambion) in a final volume of 20 μl, according to the manufacturer's instructions. DNase-treated samples were bisulphite-converted using an EZ RNA Methylation Kit (Zymo Research), following the manufacturer's

manual. As a final step before library preparation, a stepwise RNA end repair was carried out using T4 polynucleotide kinase (TaKaRa). A 3′-dephosphorylation and 5′-phosphorylation reaction was performed using T4 PNK enzyme (TaKaRa). The enzyme was removed by phenol-chloroform purification. Library preparation was done using a NEBNext Small RNA Library Prep Set, according to the manufacturer's protocol. cDNA was amplified with 12 cycles of PCR and purified using the QIAquick PCR Purification Kit (Qiagen). The libraries were size-selected on a 6% polyacrylamide gel. Compatible barcodes were selected, and samples were pooled in equimolar ratios on multiple lanes in an Illumina HiSeq 2000 platform. A 100 bp paired-end sequencing approach was used.

Bioinformatics, statistical analyses and methylation calling were performed as described in Legrand *et al* (2017), utilising the BisRNA software. Adapters were removed from sequenced reads using Cutadapt version 1.8.1 (with options: --error-rate = 0.1 --times = 2 --overlap = 1 and adapter sequences AGATCGGAAGAGCACACGTCT and GATCGTCGGACTGTAGAACTCTGAAC for forward and reverse reads, respectively (Martin, 2011). Reads were further trimmed of bases with phred quality score < 30 on 5′ and 3′ ends, and reads shorter than 25 nucleotides were discarded (Trimmomatic version 0.36) (Bolger, Lohse, & Usadel, 2014). Reads were aligned uniquely using Bsmap (version 2.87, options: -s 12 -v 0.03 -g 0 -w 1000 -S 0 -p 1 -V 1 -I 1 -n 0 -r 2 -u -m 15 -x 1000) (Xi & Li, 2009). Reference sequences were downloaded from Gtrnadb (version ce10), Ensembl (release 90, version WBcel235) and Arb-Silva (Chan & Lowe, 2016; Kersey *et al*, 2016; Lee *et al*, 2018; Quast *et al*, 2012). End sequence "CCA" was appended to tRNA if missing. Bisulphite-identical sequences, where only C>T point differences were present, were merged, keeping the C polymorphism. Similarly to Legrand *et al* (2017), tRNA sequences were further summarised to the most exhaustive yet unambiguous set of sequences, using sequence similarity matrix from Clustal Omega (Sievers *et al*, 2011). Methylation calling was performed as described in Legrand *et al* (2017), utilising the BisRNA software. Methylation frequency was calculated as the proportion of cytosines with coverage higher than 10 in three wild-type and noNSUN replicates and bisulphite non-conversion ratio higher than 0.1. The measure for reproducibility was the standard error. Deamination rates were calculated as the count of converted cytosines divided by the sum of converted and non-converted cytosines. This calculation was carried out on nuclear and mitochondrial rRNA. Known methylation sites in rRNA were removed from the calculations. WTBS raw data have been deposited in the Gene Expression Omnibus (GEO) database under the accession number GSE144822.

## Targeted bisulphite sequencing

1 µg of total RNA was bisulphite-modified with the EZ RNA Methylation Kit (Zymo Research). Briefly, samples were first treated with DNase I for 30 min at 37°C in 20 µl volume. The DNAse reaction was stopped and immediately applied to the EZ RNA Methylation Kit (Zymo Research) according to manufacturer's instructions. Converted RNAs were eluted in 12 µl of distilled water.

Reverse transcription was performed with the purified RNA, and adaptors were added to the amplicons using reverse oligonucleotides designed for the bisulphite-converted sequences of interest and SuperScript III reverse transcriptase (Invitrogen). cDNA

was cleaned from any residual RNA with an RNase H treatment at 37°C for 20 min and then used for PCR amplification and adaptor addition using forward oligonucleotides. Low annealing temperature (58°C) was used to overcome high A-T content after bisulphite treatment. 4 µl of PCR product was used for ligation and transformation into TOP10 competent cells using the Zero Blunt TOPO PCR Cloning Kit (Invitrogen) according to the manufacturer's instructions. Following overnight culture, 24 colonies were individually lysed and used for PCR amplification using M13 primers, in order to confirm the presence of the insert at the correct size by DNA electrophoresis. The remaining PCR product (10 clones per condition) was used for Sanger sequencing using T3 primers (Genewiz).

## Automated phenotypical characterisation

### Viable progeny

Viable progeny refers to the number of progeny able to reach at least the L4 stage within ~ 4 days. Measurements were completed over three 24-h intervals. First, eggs were prepared by synchronisation via coordinated egg laying. When these animals had grown to the L4 stage, single animals were transferred to fresh plates (day 0). For 3 days, each day (days 1–3), each animal was transferred to a new plate, while the eggs were left on the old plate and allowed to hatch and grow for ~ 3 days, after which, the number of animals on each of these plates was counted (Hodgkin & Barnes, 1991) using a custom animal counting program utilising short video recordings. Animals were agitated by tapping each plate four times; after this, 15 frames were imaged at 1 Hz and the maximum projection was used as a background image. Animals were then detected by movement using the difference in the image between each frame and this background image and counted this way for ten additional frames. The final count was returned as the mode of these counts. This system was tested on plates with fixed numbers of animals and was accurate to within 5%, comparable to human precision. Total viable progeny was reported then as the sum for 3 days. Data are censored for animals that crawled off of plates (Akay *et al*, 2019).

### Single worm growth curves

Populations of *Caenorhabditis elegans* were synchronised by coordinated egg laying. Single eggs were transferred to individual wells of a multi-well NGM plate solidified with Gelrite (Sigma). Each well was inoculated with 1 µl of OD 20 *E. coli* HB101 bacteria (~ 18 million) and imaged periodically using a camera mounted to a computer controlled XY plotter (EleksMaker, Jiangsu, China) which moved the camera between different wells. Images were captured every ~ 11 min for ~ 75 h. Image processing was done in real-time using custom MATLAB scripts, storing both properties of objects identified as *C. elegans*, and subimages of regions around detected objects. Body length was calculated using a custom MATLAB (Mathworks, Natick, MA) algorithm, and all other properties were measured using the *regionprops* function. Growth curves were aligned to egg-hatching time, which was manually determined for each animal.

## Polysome profiling

Synchronised populations of the strains of interest (N2 and noNSUN) were grown until adult stage (3 days) in 140 mm NGM

agar plates seeded with concentrated *E. coli* HB101 cultures at 20°C. Next, the animals were harvested from the plates, transferred to liquid cultures in S-medium supplemented with *E. coli* HB101 and incubated at 20°C or 27°C for 4 h in a shaking incubator at 200 rpm before harvesting. Sample preparation for polysome profiling was adapted from Arnold *et al* (2014). The animals were harvested, washed 3× in cold M9 buffer supplemented with 1 mM cyclohex-imide and once in lysis buffer (20 mM Tris pH 8.5, 140 mM KCl, 1.5 mM MgCl$_2$, 0.5% Nonidet P40, 2% PTE (polyoxyethylene-10-tridecylether), 1% DOC (sodium deoxycholate monohydrate), 1 mM DTT, 1 mM cycloheximide). The animals were pelleted, and as much liquid as possible was removed before the samples were frozen as droplets in liquid nitrogen, using a Pasteur pipette. Frozen droplets were transferred to metallic capsules and cryogenically ground for 25 s in a mixer (Retsch MM 400 Mixer Mill). The result-ing frozen powder was stored at −80°C until the moment of use.

Approximately 250 μl of frozen powder was added to 600 μl of lysis buffer and mixed by gentle rotation for 5 min at 4°C. The samples were centrifuged at 10,000 *g* for 7.5 min, the supernatant was transferred to fresh tubes and the RNA concentration was quantified by Nanodrop. For ribosome footprinting, 400 μl of lysate was treated with 4 μl of DNase I (1 U/μl, Thermo Scientific) and 8 μl of RNase I (100 U/μl, Ambion) for 45 min at room temperature with gentle shaking. 20 μl of RNasin ribonuclease inhibitor (40 U/μl, Promega) was added to quench the reaction when appro-priate. The tubes were immediately put on ice, and 220 μl of lysate was loaded into 17.5–50% sucrose gradients and ultracentrifuged for 2.5 h at 165,000 *gmax*, 4°C in a Beckman SW60 rotor. In paral-lel, undigested samples used for polysome profiling were equally loaded into sucrose gradients under the same conditions. Gradient fractions were eluted with an ISCO UA-6 gradient fractionator while the absorbance at 254 nm was continuously monitored. The frac-tion of polysomes engaged in translation was calculated as the area under the polysomal part of the curve divided by the area below the entire curve.

**Ribosome profiling**

Sucrose gradient fractions were collected in tubes containing 300 μl of 1 M Tris–HCl pH 7.5, 5 M NaCl, 0.5 M EDTA, 10% SDS, 42% urea and then mixed by vortexing with 300 μl of Phenol-Chloroform-Isoamylalcohol (PCL, 24:25:1). Fractions corresponding to 80S mono-somes were heated for 10 min at 65°C and centrifuged at 16,000 *g* for 20 min at room temperature. The upper aqueous phase was trans-ferred to a fresh tube and mixed well with 600 μl of isopropanol and 1 μl of Glycoblue for precipitation overnight at −80°C. RNA was pelleted by centrifugation at 16,000 *g* for 20 min at 4°C and washed in 800 μl of cold ethanol. After supernatant removal, the pellet was left to dry for 1–2 min and then dissolved in 60 μl of RNase-free water. RNA concentration and quality were measured by Nanodrop and Tapestation (2200 Agilent R6K), respectively.

A dephosphorylation reaction was performed by adding 7.5 μl T4 polynucleotide kinase (PNK) 10× buffer, 1.5 μl ATP, 1.5 μl RNase OUT, 1.5 μl T4 PNK (TaKaRa) and 3 μl RNase-free water up to a final volume of 75 μl and incubated for 1.5 h at 37°C. The enzyme was removed by acid-phenol extraction and the RNA was precipitated with 1/10 volume 3 M sodium acetate pH 5.2, 2.5× volume 100% cold ethanol and 1 μl Glycoblue overnight at −80°C.

Pelleted RNA was dissolved in RNase-free water. For footprint frag-ments purification, RNA was denatured for 3 min at 70°C and loaded into a 15% polyacrylamide TBE-urea gel alongside a small RNA marker. Gel was run for 1 h at 150 V and stained for 10 min with 1:10,000 SYBR Gold in 0.5× TBE. The gel was visualised under UV light, and the region between the 20 nt and 30 nt marks (28–32 nt) was excised with a sterile scalpel. The gel band was crushed into small pieces and incubated in 300 μl of 0.3 M RNase-free NaCl solu-tion with 2 μl of RNase OUT overnight at 4°C on an Intelli-Mixer (Elmi). The gel slurry was transferred to a 0.45 μm NanoSep MF Tube (Pall Lifesciences) and centrifuged at maximum speed for 5 min at 4°C. After overnight precipitation with 30 μl of 3 M sodium acetate pH 5.2, 1 μl Glycoblue and 800 μl 100% ethanol, the RNA was dissolved in RNase-free water.

Libraries were prepared using a NEB NEXT Small RNA Library Prep Set for Illumina (Multiplex compatible) E7330 Kit, following the manufacturer's instructions. cDNA libraries were purified according to the manual, followed by a QIAQuick PCR Purification Kit and a 6% polyacrylamide gel, where a band of 150 bp (120 bp adapter +28–32 footprint fragments) was excised. Gel extraction was performed as described above for footprint fragments purification. Libraries were sequenced at the Genomics and Proteomics Core Facility of the German Cancer Research Centre (DKFZ), Heidelberg.

Raw reads were assessed for quality using FastQC and Trimmed for low quality bases and adapter sequences using Trimmomatic (version 0.39, parameters—ILLUMINACLIP:2:30:10 SLIDINGWINDOW: 4:20 MINLEN:20) (Bolger *et al*, 2014). SortMERNA was used to remove any rRNA sequences (Kopylova *et al*, 2012). Remaining reads were uniquely aligned to the *C. elegans* (WBCel235) reference genome using HISAT2 (version 2.1.0) (Kim *et al*, 2015). The longest transcript was chosen for each gene from the WBCel235 reference genome and the CDS for these transcripts was extracted. Reads were length stratified and checked for periodicity, only read lengths show-ing periodicity over the three frames were retained for further analy-sis (26 bp −30 bp). Reads aligned to the genome were shifted 12 bp from the 5′-end towards the 3′-end (Ingolia *et al*, 2009). Any reads aligned to the first 10 codons of each gene were then removed, and the remaining reads with a 5′ end aligning to a CDS were kept for further analysis (Lecanda *et al*, 2016).

Bulk codon occupancy in the P-Site for each codon was calcu-lated as the number of shifted RPFs assigned to the first nucleotide of the codon. This value was then normalised by the frequency of the counts for the same codon in the +1, +2 and +3 codons relative to the A-Site (Stadler & Fire, 2011). Fold changes were then computed as the normalised bulk codon occupancies for noNSUN/ wild type. Ribosome occupancy for gene in a sample was calculated as the number of shifted in frame RPFs aligned to the CDS of the gene (not including the first 10 codons). These values were inputted into DESeq2 (Love *et al*, 2014). Translation efficiency was calculated by dividing the ribosome occupancy of each gene (disregarding the first 10 codons) by the mRNA abundance of the same gene. Ribo-seq raw data have been deposited in the Gene Expression Omnibus (GEO) database under the accession number GSE146256.

**RNA sequencing**

Input RNA was extracted from aliquots from the samples used for polysome profiling and ribosome footprinting. 100 μl of chloroform

were added to the tubes, which were then shaken vigorously for 15 s and incubated at room temperature for 3 min. Samples were centrifuged at 12,000 *g* for 15 min at 4°C, and the aqueous phase of the mixture was carefully recovered and transferred to a fresh RNAse-free tube. RNA was precipitated with 500 μl of cold isopropanol at room temperature for 10 min and then centrifuged at 12,000 *g* for 15 min at 4°C. The supernatant was carefully removed, the pellet was washed and vortexed with 1 ml of 75% ethanol and centrifuged at 7,500 *g* for 5 min at 4°C. RNA pellet was air-dried, dissolved in the appropriate volume of DEPC-treated water and the concentration, 260/280 and 260/230 ratios were measured by Nanodrop. RNA integrity was evaluated in the Agilent 2200 Tapestation system. RNA was depleted of DNA with a TURBO DNA-free kit (Invitrogen), according to the manufacturer's instructions. Libraries were prepared with 750 ng of starting material using the NEBNext Ultra II Directional RNA Library Prep Kit for Illumina, following rRNA depletion using a NEBNext rRNA Depletion Kit (Human/Mouse/Rat) (NEB).

Raw reads were assessed for quality using FastQC (Andrews, 2010) and Trimmed for low quality bases and adapter sequences using Trimmomatic (version 0.39, parameters—ILLUMINA-CLIP:2:30:10 SLIDINGWINDOW:4:20 MINLEN:25) (Bolger *et al*, 2014). SortMERNA (Kopylova *et al*, 2012) was used to remove any reads matching rRNA sequences. Remaining reads were aligned to the *C. elegans* reference genome (WBCel235) using HISAT2 (version 2.1.0, default parameters) (Kim *et al*, 2015). Read alignments were then counted using HTSeq-count (Anders *et al*, 2015) and gene counts inputted into DESeq2 (Love *et al*, 2014). RNA-seq raw data have been deposited in the Gene Expression Omnibus (GEO) database under the accession number GSE146256.

# Data availability

- BS-seq: Gene Expression Omnibus (GEO) GSE144822 (https://www.ncbi.nlm.nih.gov/geo/query/acc.cgi?acc=GSE144822).
- RNA-seq: Gene Expression Omnibus (GEO) GSE146256 (https://www.ncbi.nlm.nih.gov/geo/query/acc.cgi?acc=GSE146256)
- Ribo-seq: Gene Expression Omnibus (GEO) GSE146256 (https://www.ncbi.nlm.nih.gov/geo/query/acc.cgi?acc=GSE146256)

**Expanded View** for this article is available online.

## Acknowledgements

We would like to thank the Gurdon Institute Media Kitchen for their support providing reagents and media. We thank Kay Harnish for the Gurdon Institute Sequencing Facility management. We thank Julie Ahringer's laboratory for kindly sharing RNAi bacterial clones. We thank the National Bioresource Project (Tokyo, Japan) for providing the *nsun-5* deletion allele. We thank Claudia Flandoli for illustrating our working model. We thank Archana Yerra for manuscript editing. We are grateful to the Miska Laboratory members, especially Kin Man Suen, Eyal Maori, Ragini Medhi and Grégoire Vernaz, for helpful discussions and advice. We are grateful to Miranda Landgraf for her constant support and to Marc Ridyard for laboratory management and maintenance of our nematode collection. This research was supported by a Wellcome Trust Senior Investigator award (104640/Z/14/Z) and Cancer Research UK award (C13474/A27826) to E.A.M ; Conselho Nacional de Desenvolvimento Científico e Tecnológico doctorate scholarship (CNPq, Brazil—205589/2014-6) to I.C.N.; Deutsche Forschungsgemeinschaft (SPP1784) to F.L. The Miska Laboratory was also supported by core funding from the Wellcome Trust (092096/Z/10/Z, 203144/Z/16/Z) and Cancer Research UK (C6946/A24843).

## Author contributions

Conceptualisation, ICN and EAM; Investigation, ICN, FT, DJ, AGH, FB, AK and AA; Formal analysis, DJ, CL and JP; Writing—Original Draft, ICN; Writing—Review and Editing, ICN, FT, AA, FL and EAM; Supervision: MH, FL and EAM; Funding Acquisition, MH, FL and EAM.

## Conflict of interest

E.A.M. is a co-founder and director of Storm Therapeutics, Cambridge, UK. A.H. is an employee of Storm Therapeutics, Cambridge, UK.

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
