## [Review Process File · The EMBO Journal]

Translational adaptation to heat stress is mediated by RNA 5-methylcytosine in *Caenorhabditis elegans*

Isabela Navarro, Francesca Tuorto, David Jordan, Carine Legrand, Jonathan Price, Fabian Braukmann, Alan Hendrick, Alper Akay, Annika Kotter, Mark Helm, Frank Lyko, and Eric Miska
DOI: [10.15252/embj.2020105496](https://doi.org/10.15252/embj.2020105496)

Corresponding author(s): Eric Miska (eam29@cam.ac.uk)

Review Timeline:	Submission Date:	2nd May 20
	Editorial Decision:	1st Jul 20
	Revision Received:	4th Oct 20
	Editorial Decision:	30th Oct 20
	Revision Received:	4th Nov 20
	Accepted:	10th Nov 20

Editor: Stefanie Boehm

Transaction Report:

Thank you again for submitting your manuscript for consideration by The EMBO Journal and sending the pre-decision point-by-point response to the initial reviewer comments. I have now had the chance to look at your response and also discussed this with other members of the editorial team. As you appear to be in the position to address the key issues raised, we would now like to invite you to prepare and submit a revised manuscript. In this version it will be crucial to fully address the concerns raised by referee #2 as well as referee #3's points 5, 6, 7, which our external advisor also pointed out. Here, it is in particular important to resolve all issues regarding figure 5 (i.e. ref#3-point 6) and to provide a comprehensive statistical analysis and full experimental details. In addition to revising the text and expanding the discussion regarding the results at high temperature, it will also be important to revise the figure and figure legend to include more detailed information for the reader, in particular the GO term analysis in 5D (i.e. what do the axes represent?, does the size of the circles correspond to the number of genes?, are these defined gene sets?) and in the comparison of wild-type and mutant (i.e. does red indicate an upregulation in the mutant or wild-type?) in 5B.

Referee #2:

In their manuscript, Navarro and colleagues provide a global analysis of the cytosine-5 methylome in the nematode *C. elegans*. Using genome-editing to substitute key residues within the catalytic sites of the four m5C methyltransferases, they generate an organism devoid of m5C, revealing that, despite its persistence during evolution, this modification is not essential. They further exploit the potential of the m5C-null organism as an ideal control for a robust transcriptome-wide mapping of m5C sites using bisulfite sequencing. In addition to confirming the presence of m5C at various conserved sites in cytoplasmic and mitochondrial rRNAs and tRNAs in this species, they uncover novel m5C modification sites in small non-coding RNAs and their data suggest that m5C modification of mRNAs is, if present, limited and occurs at only a low level. Using single mutant organisms, they are also able to attribute specific enzymes to particular modifications, revealing that, in nematodes, NSUN4 acts as a multifunctional rRNA and tRNA methyltransferase. Ribosome profiling of m5C null worms reveals reduced translational efficiency of proline and leucine codons, consistent with the presence of m5C in the cognate tRNAs. More specifically, increased ribosome occupancy at UUG (Leu) codons during heat stress implicates the m5C present at the wobble position of tRNA^{Leu}(CAA) in cellular adaptation to temperature variation. In line with this finding, worms lacking m5C displayed defects in growth rate and progeny number, phenotypes that are exacerbated by increased temperature.

This elegant study from the Miska lab is highly interesting, and the data are very convincing and clearly presented. In contrast to studies of individual methyltransferases and their targets, the author's comprehensive approach to analysing m5C in a model organism allows them to address the long-standing conundrum of why apparently non-essential RNA modifications are conserved through evolution, and their results offer new perspectives on the functions of m5C in cellular adaptation to stress conditions. After addressing the few minor points below, the reviewer therefore strongly supports publication of this work in *The EMBO Journal*.

1. The relative proportions of m5C and f5C at position 34 of mitochondrial tRNA^{Met} in human cells has been extensively discussed with f5C being considered the predominant species. It is intriguing, therefore, that the authors detect a high level of m5C at this position in *C. elegans*. The authors suggest that this difference could arise due to preferential sequencing of immature tRNA^{Met} that has not undergone oxidative modification, however the presence of f5C₃₄ in this tRNA should be tested experimentally, either using reduced bisulfite sequencing or an alternative f5C detection approach. In case their conclusion of m5C at this position in *C. elegans* is supported, the authors should investigate and comment on the presence of ALKBH1 in this species.
2. The finding that *C. elegans* NSUN4 is multifunctional and also fulfils a role carried out by NSUN3 in higher eukaryotes indicates a divergence of its functions during evolution. It would therefore be interesting to perform a phylogenetic analysis of these proteins to determine at which stage this event arose. Related, as homologues of NSUN6, NSUN7 and DNMT2 were not identified in *C. elegans*, the authors should discuss whether the known modifications targets of these enzymes (e.g. C72 of tRNA^{Cys/Thr} and C38 of tRNA^{Asp}) are also not present in nematodes, or if, similar to NSUN4/m5C₃₄ mt-tRNA^{Met}, other NSUN proteins are responsible for installing these modifications.
3. The transcriptome-wide m5C mapping approach identified numerous cytosines in mRNAs that are resistant to bisulfite treatment. However, due to the low non-conversion rate and the high propensity for false positives, these potential modifications are strongly questioned. While this cautious approach to modification calling is commendable, before dismissing all these sites, it would be interesting to determine whether the identified mRNAs/sites have any common features that could support even a subset of them as bona fide modifications. For example, do the potential m5C-containing mRNAs encode proteins acting in a particular cellular process, or do the potential

modifications lie within sequence or structure contexts that resemble any of the known NSUN2 modifications in other RNAs (in *C. elegans* or other organisms)? Furthermore, have the authors considered these potential m5C sites in the analysis of their ribosome profiling data - i.e. do they observe increased ribosome occupancy at potentially modified sites?

4. When discussing the essential functions of rRNA methyltransferases that do not require their catalytic activity, another example that could be included is the m7G methyltransferase WBSR22 that is required for small ribosomal subunit export.

Referee #3:

The paper by Navarro and co-authors presents an inventory of the role of m5C-modification in *C. elegans* and based on deep-seq approaches combined with mass spectrometry assigns function of various NSUN methylases in nematodes. Using CRISPR-Cas9 system the authors introduce mutations at *nsun-1,2,3* rendering these methyltransferases inactive; they also produce a mutant with simultaneously depleted (total m5C) methylation activity in worms. m5C seems non-essential at permissive growth, but increases the sensitivity of the animals to thermal stress. While the breadth of the data is large, they seem contradictory in some aspects and do not support the conclusions, which decreases the enthusiasm for this paper. Specifically, these include:

1. P.4, Results section: Silencing of *nsun-1, 2, 4* and *5* was performed by RNAi, however no information on silencing level is provided. The readout to measure the effect is fairly insensitive and should be complemented with more sensitive physiological tests, including life-span assays.

2. Fig. 1E shows no significant changes in the total m5C following deletion of all *nsun-s*. Only 25% of m5C would decrease, raising the question as to where the other 75% would come from [as if a large fraction of m5C is still present despite deletion of all *nsun-s*]. Fig. 1C and D show no effect following deletion of single *nsun* genes, except *nsun2*. In stark contrast to Fig 1E, the in panels C/D show a loss of m5C following all *nsun*-depletion. Is the readout sensitivity sufficient enough to support authors' claims here?

3. P.5, first sentences in the paragraph: 'm5C is abundant in tRNA, mt-tRNA...' is unsubstantiated. Actually, the authors have the tools for this and should reveal (using m5C-sequencing along with lifespan analysis) the reason for the differences. As a general comment for the whole study, in the majority of the experiments a readout that provides a single snapshot of a physiological stage is used. The m5C might have more kinetic-life effect, so that readouts (with time points), e.g. lifespan analysis should be used to assess the effect of *nsun* single-gene and all-genes depletion.

4. The title and the abstract do not recapitulate the findings in the paper - both are written in a quite general and unprecise way and overstated. It reads that m5C is a prevalent modification in all RNA species, and only half a way down in the manuscript to read that majorly two rRNA and few tRNA modifications are meant.

5. Which *nsun* is then responsible for the rRNA modifications? (Fig. 2A). Since rRNA is the most abundant species among RNAs, wouldn't be seen in the total m5C measurements (Fig. 1C,D) following depletion of single *nsun* genes?

6. The subsection on the decoding efficiency of Leu/Pro codons is difficult to follow and this is the section of the manuscript where the conclusions are absolutely not substantiated by the data. It is hard to agree with this section. No colorcode is present on Fig. 2B so that the position of the Leu or Pro tRNA isoacceptors cannot be seen on the plot. The UUG codon read tRNA(Leu)CAA shows a marginal change in the ribosome occupancy, but none of the Pro-codons as opposed to the claims in the text. On the contrary, Pro codons at high temperature should be read even faster, as the ribosome occupancy is lower/decreasing. Is the occupancy seen for Leu-UUG codon significant? It seems less than one-fold, i.e. less than 2x. Whether this is an effect to significantly change

ribosome traffic and consequently gene expression is questionable.

7. The comparisons of the Pro/Leu codon-enriched genes in their current version are meaningless, but also incorrect to draw conclusions based on such analysis. First, it seems that all Leu/Pro codons were considered, though not all of them are read by m5C-tRNAs. Second, Pro is relatively abundant codon which increases the false-positives. Third, any large group of genes (because of two) will then deliver mathematically a significance value, which may not have biological meaning. The authors should look for only Leu-UUG as this is the only one with increased ribosomal occupancy, if the value proves to be significant. Also, should provide evidence with example genes in which the Leu-UUG occupancy is indeed above the average of the other codons. Fig. 5B- is this showing A or P-site occupancy?

8. Polysome profiling is mentioned in the methods section, but no results on this are shown. Those could contain crucial information for the polysome:monosome ratios among the mutants.

Point-by-point response to the reviewers' comments

Reviewer comments: in blue font

Point-by-point response of the authors: in black font

Referee #2 (Report for Author)

In their manuscript, Navarro and colleagues provide a global analysis of the cytosine-5 methylome in the nematode *C. elegans*. Using genome-editing to substitute key residues within the catalytic sites of the four m5C methyltransferases, they generate an organism devoid of m5C, revealing that, despite its persistence during evolution, this modification is not essential. They further exploit the potential of the m5C-null organism as an ideal control for a robust transcriptome-wide mapping of m5C sites using bisulfite sequencing. In addition to confirming the presence of m5C at various conserved sites in cytoplasmic and mitochondrial rRNAs and tRNAs in this species, they uncover novel m5C modification sites in small non-coding RNAs and their data suggest that m5C modification of mRNAs is, if present, limited and occurs at only a low level. Using single mutant organisms, they are also able to attribute specific enzymes to particular modifications, revealing that, in nematodes, NSUN4 acts as a multifunctional rRNA and tRNA methyltransferase. Ribosome profiling of m5C null worms reveals reduced translational efficiency of proline and leucine codons, consistent with the presence of m5C in the cognate tRNAs. More specifically, increased ribosome occupancy at UUG (Leu) codons during heat stress implicates the m5C present at the wobble position of tRNA^{Leu}(CAA) in cellular adaptation to temperature variation. In line with this finding, worms lacking m5C displayed defects in growth rate and progeny number, phenotypes that are exacerbated by increased temperature.

This elegant study from the Miska lab is highly interesting, and the data are very convincing and clearly presented. In contrast to studies of individual methyltransferases and their targets, the author's comprehensive approach to analysing m5C in a model organism allows them to address the long-standing conundrum of why apparently non-essential RNA modifications are conserved through evolution, and their results offer new perspectives on the functions of m5C in cellular adaptation to stress conditions. After addressing the few minor points below, the reviewer therefore strongly supports publication of this work in *The EMBO Journal*.

Thank you. We appreciate the valuable and constructive comments.

1. The relative proportions of m⁵C and f⁵C at position 34 of mitochondrial tRNA^{Met} in human cells has been extensively discussed with f⁵C being considered the predominant species. It is intriguing, therefore, that the authors detect a high level of m⁵C at this position in *C. elegans*. The authors suggest that this difference could arise due to preferential sequencing of immature tRNA^{Met} that has not undergone oxidative modification, however the presence of f⁵C₃₄ in this tRNA should be tested experimentally, either using reduced bisulfite sequencing or an alternative f⁵C detection approach. In case their conclusion of m⁵C at this position in *C. elegans* is supported, the authors should investigate and comment on the presence of ALKBH1 in this species.

Thank you, we appreciate this very relevant comment and have taken this into consideration during the development of this project. Indeed, it is surprising that we detect m⁵C at such high levels in this position. Our suggestion that this might result from a methodological bias arises from the fact that a previously published paper has elegantly demonstrated the presence of f⁵C in *C. elegans* mt-tRNA Met-CAU (Nakano *et al*, 2016, Supplementary Figure 4 attached below for ease of reference, doi: 10.1038/nchembio.2099). This work specifically isolated mt-tRNA Met-CAU using DNA probes in reciprocal circulating chromatography and subjected it to mass spectrometry measurements, demonstrating high stoichiometry of f⁵C in these molecules. This technique is the gold standard for this type of analysis and therefore we considered unnecessary to reproduce this finding in our study. We included a more thorough discussion of this point in the revised version of the manuscript, mentioning also the presence of an ALKBH1 homologue in the nematode (Y51H7C.5) and its more recently discovered involvement in mitochondrial protein biogenesis (lines 377-382).

Supplementary Figure 4. Presence of f⁵C₃₄ in *C. elegans* mt tRNA^{Met}

(a) Mass chromatograms of the sextuply charged negative ions of the anticodon-containing RNase T₁ fragment of *C. elegans* mt tRNA^{Met}. Top, middle, and bottom panels indicate fragments possessing f⁵C, m⁵C and unmodified C at position 34, respectively. Charge state (z) of each m/z value is parenthesized.

(b) CID spectrum of the f⁵C-containing fragment. Product ions are assigned in the spectrum.

2. The finding that *C. elegans* NSUN4 is multifunctional and also fulfils a role carried out by NSUN3 in higher eukaryotes indicates a divergence of its functions during evolution. It would therefore be interesting to perform a phylogenetic analysis of these proteins to determine at which stage this event arose. Related, as homologues of NSUN6, NSUN7 and DNMT2 were not identified in *C. elegans*, the authors should discuss whether the known modifications targets of these enzymes (e.g. C72 of tRNACys/Thr and C38 of tRNAAsp) are also not present in nematodes, or if, similar to NSUN4/m5C34 mt-tRNAMet, other NSUN proteins are responsible for installing these modifications.

Thank you for these interesting suggestions. To address the first point, we performed a phylogenetic analysis of NSUN3 using Treefam and, due to the high similarity, NSUN4 sequences were automatically included in the resulting cladogram. While *Drosophila* and *C. elegans* only have NSUN4, vertebrate model organisms as basal as zebrafish have both NSUN3 and NSUN4 (Figure 3E, lines 245-252). A more expanded tree indicates the presence of NSUN4, but not NSUN3, in sea lampreys, suggesting that NSUN3 might have diverged from NSUN4 in vertebrates (<http://www.treefam.org/family/TF321304#tabview=tab1> – “full” tree type).

To address the second point, we included the methylation status of targets sites of NSUN6 and DNMT2 in Figure 2C (lines 191-193), demonstrating that these sites are not methylated in the nematode. We were not able to identify homologous sequences of Pfk1 and Sirt5 eRNAs, which have been shown to be methylated by NSUN7.

3. The transcriptome-wide m5C mapping approach identified numerous cytosines in mRNAs that are resistant to bisulfite treatment. However, due to the low non-conversion rate and the high propensity for false positives, these potential modifications are strongly questioned. While this cautious approach to modification calling is commendable, before dismissing all these sites, it would be interesting to determine whether the identified mRNAs/sites have any common features that could support even a subset of them as bona fide modifications. For example, do the potential m5C-containing mRNAs encode proteins acting in a particular cellular process, or do the potential modifications lie within sequence or structure contexts that resemble any of the known NSUN2 modifications in other RNAs (in *C. elegans* or other organisms)? Furthermore, have the authors considered these potential m5C sites in the analysis of their ribosome profiling data - i.e. do they observe increased ribosome occupancy at potentially modified sites?

Thank you for these suggestions. To address this point, we selected 188 sites that remained 25-40% unconverted after sodium bisulfite treatment exclusively in wild type samples and performed gene ontology, motif search, genomic localisation and secondary structure analyses on these transcripts, as well as localisation analyses of each site within the transcripts. In addition, we investigated the expression and translation of these transcripts in our ribosome profiling data, however no significant or shared features were found (lines 204-208).

4. When discussing the essential functions of rRNA methyltransferases that do not require their catalytic activity, another example that could be included is the m7G methyltransferase WBSCR22 that is required for small ribosomal subunit export.

Thank you for pointing out to this relevant reference. In the revised version of the manuscript we included WBSCR22 to our discussion about catalysis-independent functions of RNA methyltransferases (lines 352-356).

Referee #3 (Report for Author)

The paper by Navarro and co-authors presents an inventory of the role of m5C-modification in *C. elegans* and based on deep-seq approaches combined with mass spectrometry assigns function of various NSUN methylases in nematodes. Using CRISPR-Cas9 system the authors introduce mutations at *nsun-1,2,3* rendering these methyltransferases inactive; they also produce a mutant with simultaneously depleted (total m5C) methylation activity in worms. m5C seems non-essential at permissive growth, but increases the sensitivity of the animals to thermal stress. While the breadth of the data is large, they seem contradictory in some aspects and do not support the conclusions, which decreases the enthusiasm for this paper. Specifically, these include:

1. P.4, Results section: Silencing of *nsun-1, 2, 4* and *5* was performed by RNAi, however no information on silencing level is provided. The readout to measure the effect is fairly insensitive and should be complemented with more sensitive physiological tests, including life-span assays.

While we appreciate that alternative physiological tests could have been performed to evaluate the effect of silencing of *nsun* genes, the scope of our manuscript is the study of m⁵C itself, rather than other, potentially catalysis-independent, functions of m⁵C RNA methyltransferases. Of relevance to this comment is the preprint recently published by Heissenberger *et al.*, showing that whole-animal or germline-specific *nsun-1* RNAi during adulthood in *C. elegans* has no effect on animal lifespan, but depletion of *nsun-1* in somatic tissues increases mean lifespan by ~10% (doi: 10.1101/2020.03.16.993469). This is in contrast with the previously observed extension of lifespan by 17% following whole-animal *nsun-5* knockout (Schosserer *et al.*, 2015, doi: 10.1038/ncomms7158). The referred preprint also confirms our finding that *nsun-1* silencing from the first larval stages leads to complete infertility upon adulthood, and further narrows down this phenotype to a soma-dependent defect in oocyte maturation (Heissenberger *et al.*, 2020, doi: 10.1101/2020.03.16.993469).

2. Fig. 1E shows no significant changes in the total m5C following deletion of all *nsun*-s. Only 25% of m5C would decrease, raising the question as to where the other 75% would come from

[as if a large fraction of m⁵C is still present despite deletion of all nsun-s]. Fig. 1C and D show no effect following deletion of single nsun genes, except nsun2. In stark contrast to Fig 1E, the in panels C/D show a loss of m⁵C following all nsun-depletion. Is the readout sensitivity sufficient enough to support authors' claims here?

Figure 1E does not present any measurements of m⁵C levels. We believe this comment refers to the levels of 3-methylcytosine presented in Figure 1E (Figure 1G in the revised version), a chemically-unrelated modification which, as expected, does not significantly change upon loss of m⁵C RNA methyltransferases in the noNSUN strain. m⁵C and hm⁵Cm measurements were performed using mass spectrometry, the gold standard technique for quantification and sensitivity in the field, and are presented in Figures 1E-F of the revised manuscript. As clearly stated in the text, our analyses have a lower limit of detection of ~0.3 ng/ml and m⁵C has consistently not been detected in the noNSUN strain in any of the numerous measurements performed using distinct equipment, several of which were not included in this manuscript. Therefore, the readout sensitivity is sufficient to support our claims. Further clarification regarding the lack of clear reduction of m⁵C levels upon loss of NSUN-1 and NSUN-5 activity is provided in point 5 below.

3. P.5, first sentences in the paragraph: 'm⁵C is abundant in tRNA, mt-tRNA...' is unsubstantiated. Actually, the authors have the tools for this and should reveal (using m⁵C-sequencing along with lifespan analysis) the reason for the differences. As a general comment for the whole study, in the majority of the experiments a readout that provides a single snapshot of a physiological stage is used. The m⁵C might have more kinetic-life effect, so that readouts (with time points), e.g lifespan analysis should be used to assess the effect of nsun single-gene and all-genes depletion.

Thank you for this observation. We agree that the statement “m⁵C is abundant in tRNA, mt-tRNA, rRNA, ncRNA but not mRNA in *C. elegans*” is not accurate. It does not represent the message we would like to convey in the paper and it has been corrected in the revised version of the manuscript (line 166).

While we can appreciate that most of our m⁵C measurements have focused on a specific life stage of *C. elegans*, we did perform sensitive phenotypical analyses of development and fertility across the different strains, showing the effect of m⁵C loss in specific stages, e.g. a marked difference in body length during the transition from L4 stage to young adulthood (Figure 4A). We agree that m⁵C might be dynamic and have different levels/effects throughout development, however we do not see how lifespan analyses specifically would increase our understanding of this matter. This could be an interesting avenue for future projects, however it is not in the scope of the present manuscript.

4. The title and the abstract do not recapitulate the findings in the paper - both are written in a quite general and unprecise way and overstated. It reads that m⁵C is a prevalent modification

in all RNA species, and only half a way down in the manuscript to read that majorly two rRNA and few tRNA modifications are meant.

We do not state that m⁵C is a prevalent modification in all RNA species in the title, nor in the abstract. In fact, we do not make such a statement at all throughout the whole manuscript. In the abstract it reads “Methylation of carbon-5 of cytosines (m⁵C) is a post-transcriptional nucleotide modification of RNA found in all kingdoms of life.”, a statement mainly supported by a paper published by ours and the Balasubramanian lab (Huber *et al*, 2015, doi: 10.1002/cbic.201500013), and referring to the degree of evolutionary conservation of the modification, rather than its distribution or abundance.

5. Which nsun is then responsible for the rRNA modifications? (Fig. 2A). Since rRNA is the most abundant species among RNAs, wouldn't be seen in the total m⁵C measurements (Fig. 1C,D) following depletion of single nsun genes?

The enzymatic specificity of rRNA methylation sites is demonstrated by targeted bisulfite sequencing data presented on Figure EV1 of the revised manuscript (former Supp Figure 2). Labelling has been improved in this figure to facilitate interpretation.

Our data presented on Figure EV1A clearly shows that position C2381 is a target of NSUN-5, which is in agreement with other publications (Schosserer *et al*, 2015, doi: 10.1038/ncomms7158; Heissenberger *et al.*, 2020, doi: 10.1101/2020.03.16.993469).

The data presented on Figure EV1B shows that methylation of C2982 is the product NSUN-1 activity. The sequenced clones behave as expected, *i.e.* non-converted C2982 position in *nsun-1* mutants, converted C2982 position in *nsun-5* and wild type strains. In addition, this site is a conserved *nop2* (yeast NSUN-1 homologue) target in *Saccharomyces cerevisiae* (please see alignment below and Sharma *et al*, 2013, doi: 10.1093/nar/gkt679) and has been recently confirmed as a target of NSUN-1 in *C. elegans* in a recently published preprint (Heissenberger *et al.*, 2020, doi: 10.1101/2020.03.16.993469)

Regarding the lack of a clear reduction in m⁵C levels following NSUN-1 and NSUN-5 individual mutations, it is worth noting that, according to our BS-Seq analyses and to the available literature, these enzymes modify single positions in rRNA and in reduced stoichiometry in comparison to NSUN-2, which modifies 40 different positions in tRNAs often to stoichiometry close to 100% (at least 50% in our dataset). Upon loss of NSUN-2 activity, it is still possible to detect 15% of m⁵C levels, which represents the contribution of NSUN-1,

NSUN-4 and NSUN-5 in our samples. In contrast, levels of hm⁵Cm, a much rarer modification thus far known to only occur in NSUN-2 targets, are completely abolished upon loss of this enzyme, demonstrating the accuracy of our measurements.

6. The subsection on the decoding efficiency of Leu/Pro codons is difficult to follow and this is the section of the manuscript where the conclusions are absolutely not substantiated by the data. It is hard to agree with this section. No colorcode is present on Fig. 2B so that the position of the Leu or Pro tRNA isoacceptors cannot be seen on the plot. The UUG codon read tRNA(Leu)CAA shows a marginal change in the ribosome occupancy, but none of the Pro-codons as opposed to the claims in the text. On the contrary, Pro codons at high temperature should be read even faster, as the ribosome occupancy is lower/decreasing. Is the occupancy seen for Leu-UUG codon significant? It seems less than one-fold, i.e. less than 2x. Whether this is an effect to significantly change ribosome traffic and consequently gene expression is questionable.

We chose not to use a colour code on Figure 2B due to the large amount of data presented. Rather, we chose to present individual non-conversion rates for all the positions showing >50% non-conversion on Figure 2C, where tRNAs are organised in alphabetical order to facilitate the identification of the tRNA position of the reader's interest.

We acknowledge that the differences observed in proline codon occupancy do not occur in a heat shock-dependent manner. In our view, UUG, leucine and proline translation are distinctly affected upon loss of m⁵C. UUG translation is the most strongly affected, specifically upon heat shock, which suggests an involvement of m⁵C wobble methylation in translation under temperature stress. On the other hand, leucine translation efficiency is affected in both temperatures tested and proline exclusively at 20°C. We re-wrote this section to better convey our message and clarify any perceived contradictions (lines 298-318).

As indicated on the figure legend, codon occupancy is presented on Figure 5B as log₂ of the fold change, *i.e.* the average fold change observed in UUG codon occupancy is 1.75. While this may seem like a small difference, it is worth noting that such measurements are performed at genomic scale. Other publications have presented similar or subtler differences in codon occupancy and still were able to detect significant effects in the proteome as a consequence. For examples, please refer to Tuorto *et al*, 2018 (doi: 10.15252/embj.201899777) and Nedialkova *et al*, 2015 (doi: 10.1016/j.cell.2015.05.022). In the revised version of the manuscript we have included statistical analyses demonstrating that not only UUG, but other triplets show significantly increased ribosome occupancy at 20 or 27°C (p-value < 0.05). These triplets, with one exception, encode leucine and proline (Figure EV4A, lines 298-318).

7. The comparisons of the Pro/Leu codon-enriched genes in their current version are meaningless, but also incorrect to draw conclusions based on such analysis. First, it seems that all Leu/Pro codons were considered, though not all of them are read by m⁵C-tRNAs. Second,

Pro is relatively abundant codon which increases the false-positives. Third, any large group of genes (because of two) will then deliver mathematically a significance value, which may not have biological meaning. The authors should look for only Leu-UUG as this is the only one with increased ribosomal occupancy, if the value proves to be significant. Also, should provide evidence with example genes in which the Leu-UUG occupancy is indeed above the average of the other codons. Fig. 5B- is this showing A or P-site occupancy?

We do not agree with this interpretation. First, our BS-seq data presented on Figure 2C demonstrates that all leucine (AAG, CCG, UCG, CCU and CUC) and proline (AGG, CGG, UGG) tRNAs encoded in *C. elegans* genome are methylated in at least one position (<http://gtrnadb.ucsc.edu/genomes/eukaryota/Celeg11/>). Second, it is not clear to us how the fact that proline is a relatively abundant amino acid selectively invalidates our findings on these codons, as leucine is in fact the most frequent amino acid in *C. elegans* (doi: 10.1016/S0168-9525(00)02041-2). Third, it is true that a statistically significant change does not necessarily translate into biological meaning, however this is true for any statistical test. In this case, we have performed a t-test with all underlying assumptions being met and we use the results to explain that the translation efficiency of genes enriched in specific codons is reduced upon loss of m⁵C.

We thank the suggestion regarding UUG-specific analyses. In the revised manuscript, we provide representative examples of CDS where ribosome stalling occurs at UUG codons in the noNSUN strain upon heat shock. Interestingly, this effect seems to be context-dependent, as it is not observed in every UUG codon within a given CDS (Figure 5C, Figure EV4B). Following statistical confirmation that UUG codons show significantly increased ribosome occupancy (please refer to point 6), we performed translation efficiency analyses focusing on UUG-enriched genes and showed that these transcripts are less efficiently translated in noNSUN samples upon heat shock. This set of genes is not enriched in any specific biological process (Figure 5D-E, lines 298-318).

Finally, in Figure 5B we show P-site occupancy, and this information has been added to the figure legend in the revised manuscript.

8. Polysome profiling is mentioned in the methods section, but no results on this are shown. Those could contain crucial information for the polysome:monosome ratios among the mutants.

Polysomal fractions measured by polysome profiling in wild type and noNSUN strains at different temperatures are presented on Figure 5A. No differences were found in both conditions tested.

Thank you for submitting your revised manuscript. We have now received the reports from one of the initial referees, as well as the advisor we had contacted (now referee #4), please see their comments below. I am happy to say that they overall find that the issues have been satisfactorily addressed and now support publication. Referee #4 raises some remaining points, which should be clarified in the final revised version. Please also provide a brief point-by-point response to these comments when submitting the manuscript. In addition, I would also like to ask you to address a number of editorial issues that are listed in detail below. Please make any changes to the manuscript text in the attached document only using the "track changes" option. Once these remaining issues are resolved, we will be happy to formally accept the manuscript for publication.

REFeree REPORTS

Referee #2:

This manuscript by Navarro and colleagues describing transcriptome-wide analysis of 5-methylcytosine and the methyltransferases responsible for installing it in *C. elegans* is a robust and interesting study. The authors have systematically and satisfactorily addressed my questions, and I recommend publication of the revised manuscript in the EMBO Journal.

Referee #4:

The authors have addressed the reviewers' comments to my satisfaction. I support publication with the following minor modifications and clarifications:

1. Abstract, l 44: "... loss of m5C generally impacts decoding of these two amino acids..." - this claim seems exaggerated given that the examples clearly show, and the authors state in the text, that

only a (small) subset of codons appears affected; please rephrase. Similarly, l. 307, "... as it is not observed in every UUG" - looking at Fig. 5C, EV4B, it would seem more appropriate to state "as it appears to occur in only a small subset of UUG"

2. Fig. 1C: I am confused about the circles - why are there three if two independent experiments only? What exactly are three "biological replicates" in these independent experiments - merely separate plates from the same egg prep?

3. Ref. 3, point 5/Fig. EV1B: I find the assignment of C2982 to NSUN-1 rather poorly documented given only one read each in nsun-1 and nsun-5 mutants (and three in wt). This is not a major point in the paper, but the authors might want to point the reader to the preprint that they cite in support of their case in the EV1B figure legend.

4. L. 152: please put the lower limit of detection in perspective, how does this compare to what is seen in wild-type samples.

5. Fig. 3A, C, D: Although a number of biological replicates is given, it is not clear how this is used in this analysis, it does not permit any conclusion on reproducibility since everything is lumped together. Some clarifying statement (e.g. "similar effects were seen in each of these replicates") would help.

6. Fig. 4A: Do the animals end up small or are they Dpy? Has it been confirmed that developmental rates are indeed the same, or is it possible that nsun mutant animals are simply not yet adult at the time wild-type animals are? Some clarification would help interpretation of the phenotype.

7. Fig. 5A: I would appreciate seeing the profiles in a supplementary figure, the quantification shown is fairly derived.

8. Fig. 5D: please clarify in the legend how enrichments for "UUG enriched" etc. were calculated/defined.

9. Fig. EV3B: are these RPFs or RPFs normalized to RNA levels (i.e, TE)?

10. Personally, I find Fig. 5E and related text (ll. 319 - 321) rather uninformative and unnecessary and would encourage the authors, at their discretion, to omit both. Minimally, information on what was used as background for the analysis is required.

Point-by-point response to the reviewers' comments

Reviewer comments: in blue font

Point-by-point response of the authors: in black font

1. Abstract, l 44: "... loss of m5C generally impacts decoding of these two amino acids..." - this claim seems exaggerated given that the examples clearly show, and the authors state in the text, that only a (small) subset of codons appears affected; please rephrase. Similarly, l. 307, "... as it is not observed in every UUG" - looking at Fig. 5C, EV4B, it would seem more appropriate to state "as it appears to occur in only a small subset of UUG"

Thank you for this observation, we agree that these suggestions make the presentation of our findings more precise and have edited the text to reflect the changes.

2. Fig. 1C: I am confused about the circles - why are there three if two independent experiments only? What exactly are three "biological replicates" in these independent experiments - merely separate plates from the same egg prep?

We apologise for the lack of clarity. Here biological replicates refer to separate plates, *i.e.* this experiment was repeated twice with three RNAi plates per condition tested each time. In all plates analysed we observed 100% of the adults being infertile upon *nsun-1* knockdown and 100% of the adults being fertile upon knockdown of the other *nsun* genes. The legend of Fig 1 has been edited for clarification.

3. Ref. 3, point 5/Fig. EV1B: I find the assignment of C2982 to NSUN-1 rather poorly documented given only one read each in *nsun-1* and *nsun-5* mutants (and three in wt). This is not a major point in the paper, but the authors might want to point the reader to the preprint that they cite in support of their case in the EV1B figure legend.

Thank you for this suggestion. Unfortunately, we encountered some technical difficulties during the cloning of amplicons of *nsun-1* target sites. As presented in the figure, only few

colonies contained the whole insert to be sequenced, and those behaved as expected. Despite that, the degree of conservation of the methylation of this site, as observed in yeast, as well as the data presented on the mentioned preprint gave us the confidence that the data is representative of reality. The legend of Figure EV1, as well as the manuscript text, have been edited to cite the referred publication.

4. L. 152: please put the lower limit of detection in perspective, how does this compare to what is seen in wild-type samples.

Thank you for this suggestion, this information has been added to the manuscript.

5. Fig. 3A, C, D: Although a number of biological replicates is given, it is not clear how this is used in this analysis, it does not permit any conclusion on reproducibility since everything is lumped together. Some clarifying statement (e.g. "similar effects were seen in each of these replicates") would help.

We apologise for the lack of clarity. In Figure 3A we show a representative plot of one of the triplicates. In Figures 3C and 3D the average of two independent experiments is presented, *i.e.* 20 sequenced clones in total. Figure 3A legend has been edited to include the relevant information and statement.

6. Fig. 4A: Do the animals end up small or are they Dpy? Has it been confirmed that developmental rates are indeed the same, or is it possible that *nsun* mutant animals are simply not yet adult at the time wild-type animals are? Some clarification would help interpretation of the phenotype.

According to our observations, there is a developmental delay in the noNSUN strain, which can be seen both as reduced body size throughout development (shown in Fig 4A) and increased time to first egg laid in comparison to the wild type strain (routine observation). In addition to that, noNSUN animals remain 20% smaller even when they reach adulthood themselves, although morphology is not grossly affected (*i.e.* animals are not Dpy). We agree that this information would help interpretation and thank the suggestion. The relevant data supporting the statement regarding body size has now been added to the manuscript in Fig 4B.

7. Fig. 5A: I would appreciate seeing the profiles in a supplementary figure, the quantification shown is fairly derived.

Polysome profiles have been included as Supplementary Figure S2.

8. Fig. 5D: please clarify in the legend how enrichments for "UUG enriched" etc. were calculated/defined.

A gene was considered enriched in a certain codon when the proportion of this codon in the gene was at least 3-fold higher than the proportion of the same codon across the transcriptome. This information has been added to the figure legend.

9. Fig. EV3B: are these RPFs or RPFs normalized to RNA levels (i.e, TE)?

These are RPFs, not translation efficiency.

10. Personally, I find Fig. 5E and related text (ll. 319 - 321) rather uninformative and unnecessary and would encourage the authors, at their discretion, to omit both. Minimally, information on what was used as background for the analysis is required.

Fig 5E and related text have been removed from the manuscript.

Thank you again for submitting the final revised version of your manuscript. I am pleased to inform you that we have now accepted it for publication in The EMBO Journal.

Corresponding Author Name: Eric A Miska

Manuscript Number: EMBOJ-2020-105496